# Robust LLM Alignment via Distributionally Robust Direct Preference Optimization

**Zaiyan Xu**[1], **Sushil Vemuri**[1], **Kishan Panaganti**[2,†], **Dileep Kalathil**[1],
**Rahul Jain**[3], **Deepak Ramachandran**[3]

[1]Texas A&M University, [2]Tencent AI Lab, [3]Google DeepMind.
Emails: {zxu43, sushil22, dileep.kalathil}@tamu.edu, kpb.research@gmail.com,
{rahulajain, ramachandrand}@google.com

## Abstract

A major challenge in aligning large language models (LLMs) with human preferences is the issue of *distribution shift*. LLM alignment algorithms rely on static preference datasets, assuming that they accurately represent real-world user preferences. However, user preferences vary significantly across geographical regions, demographics, linguistic patterns, and evolving cultural trends. This preference distribution shift leads to catastrophic alignment failures in many real-world applications. We address this problem using the principled framework of distributionally robust optimization, and develop two novel distributionally robust direct preference optimization (DPO) algorithms, namely, Wasserstein DPO (WDPO) and Kullback–Leibler DPO (KLDPO). We characterize the sample complexity of learning the optimal policy parameters for WDPO and KLDPO. Moreover, we propose scalable gradient descent-style learning algorithms by developing suitable approximations for the challenging minimax loss functions of WDPO and KLDPO. Our empirical experiments using benchmark data sets and LLMs demonstrate the superior performance of WDPO and KLDPO in substantially improving the alignment when there is a preference distribution shift.

## 1 Introduction

The alignment of large language models (LLMs) with human values and preferences is a central objective in machine learning, enabling these models to produce outputs that are useful, safe, and aligned with human intent. Since LLMs are trained on vast, diverse datasets using self-supervised learning, an additional alignment phase is often required to refine their behavior based on human feedback. A widely adopted approach for this is Reinforcement Learning from Human Feedback (RLHF) (Christiano et al., 2017; Ziegler et al., 2019; Ouyang et al., 2022), which involves training a reward model using human preference data and optimizing the LLM using reinforcement learning (RL) approaches, such as proximal policy optimization. More recently, Direct Preference Optimization (DPO) has emerged as an alternative that simplifies the alignment process by directly optimizing model parameters based on human preferences without requiring an explicit reward model. These alignment techniques have played a crucial role in improving the ability of LLMs to generate responses that adhere to human expectations and societal norms, leading to today's powerful chat models (Achiam et al., 2023; Touvron et al., 2023).

Despite the importance of the LLM alignment problem, RLHF and DPO remain fundamentally challenging and fragile, mainly due to three reasons. ($i$) *Diversity of human preferences:* Standard RLHF/DPO approaches implicitly assume that human preferences can be accurately captured by a single reward function. In reality, human preferences are highly diverse, context-dependent, and distributional, making it infeasible to represent them with a one-size-fits-all optimization framework

---

[†]Work done as postdoctoral researcher at the California Institute of Technology.

(Zhao et al., 2024; Durmus et al., 2024). Standard preference-learning methods tend to skew toward the preferences represented in the majority of training data, disproportionately penalizing minority opinions and reinforcing biases (Chakraborty et al., 2024). (*ii*) *Reward hacking:* The quality of human preference feedback is inherently noisy, ambiguous, and inconsistent, as they are collected from human annotators who may lack domain expertise, exhibit labeling fatigue, or hold conflicting opinions (Zhang et al., 2025; Wu et al., 2025), which can often lead to misaligned preference estimation. This issue is exacerbated by reward hacking, where models learn undesirable shortcuts to maximize the estimated reward function, generating responses that appear aligned but deviate from genuine human intent (Amodei et al., 2016; Skalse et al., 2022; Eisenstein et al., 2024). (*iii*) *Distribution shift:* Alignment algorithms use static preference datasets for training, collected under controlled conditions. However, the preferences of real-world users can often be out-of-distribution from that of the training data, depending on the geographical region, demography, linguistic patterns, and emerging social trends, among many others. A model aligned using a specific fixed dataset may fail catastrophically when deployed to users whose preference distribution does not match that of the training data (Casper et al., 2023; LeVine et al., 2023; Kirk et al., 2024).

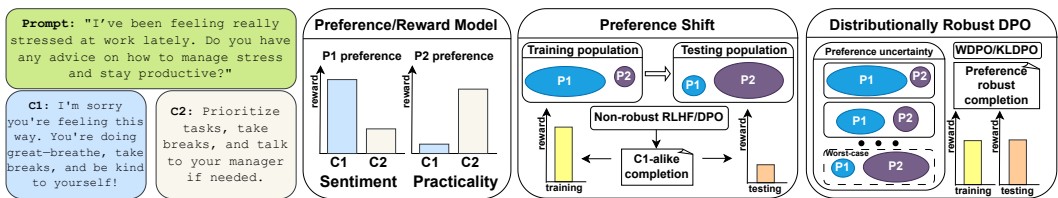

Figure 1: If the training population predominantly uses preference model 1 (P1), a non-robust RLHF/DPO model will favor Completion 1 (C1). However, deploying this model to a test population that prefers model 2 (P2), which favors Completion 2, leads to poor performance. Our distributionally robust DPO (WDPO/KLDPO) addresses this by optimizing across an uncertainty set of preference models, ensuring robust performance under preference shifts.

In this paper, we address the fragility of the LLM alignment using DPO, with a particular focus on the challenges arising from the *preference distribution shift*. DPO reduces the alignment problem to a supervised learning problem. It is known that the performance of supervised learning algorithms degrades significantly in the out-of-distribution setting (Taori et al., 2020; Koh et al., 2021), which is exacerbated due to the realistic distribution shift scenarios arising in the LLM deployment. Distributionally robust optimization/learning framework has been recently used to address the issue of distribution shift in various settings (Duchi and Namkoong, 2021; Kuhn et al., 2019; Chen et al., 2020). This framework considers an uncertainty set of data distributions around a nominal distribution (typically the training data distribution) and solves a minimax optimization problem to minimize the expected loss, where the expectation is taken with respect to the distribution in the uncertainty set that maximizes the loss. The distributionally robust learning approach has been successfully applied, with theoretical guarantees and scalable algorithms, in supervised learning (Chen and Paschalidis, 2018; Namkoong and Duchi, 2016; Levy et al., 2020), multi-armed bandits (Si et al., 2020; Yang et al., 2023), and reinforcement learning (Wang and Zou, 2022; Panaganti et al., 2022; Zhou et al., 2024; Xu et al., 2023). This motivates us to address the following questions:

> *Can distributionally robust learning mitigate the impact of distribution shift in DPO-based LLM alignment? What theoretical guarantees can be established for such methods? How can we design tractable, gradient-based algorithms to implement them? How do these approaches empirically improve alignment performance?*

Distributionally robust learning for LLM alignment presents challenges beyond standard supervised settings. In supervised learning, distributional robustness is often tractable due to well-behaved convex losses. In contrast, RL poses more complex forms of distribution shift, both exogenous (e.g., user preference drift) and endogenous (e.g., mismatch between the learned and logging policies). Although DPO is framed as a supervised objective, its likelihood-ratio formulation, based on pairwise comparisons and a reference policy, derives from KL-regularized reward maximization, linking it closely to RL. Like imitation learning and offline RL, which address RL problems through supervised proxies (e.g., behavior cloning, Q-function regression), DPO inherits the same instability and sensitivity to distribution shift. These challenges are amplified in the distributionally robust

setting, where the non-convex min-max objective is especially hard to optimize at LLM scale, making standard alternating-gradient methods unstable and impractical. We answer the above questions affirmatively and address the associated challenges through the following contributions:

1. To the best of our knowledge, this is the first work to propose a unified mathematical and algorithmic framework for addressing preference shift in LLM alignment through distributionally robust optimization. Our formulation leads to two robust DPO variants, Wasserstein DPO (WDPO) and Kullback-Leibler DPO (KLDPO), with provable guarantees. In particular, for log-linear policies, we show that the estimation error of the robust policy parameters converges at a rate of $O(n^{-1/4})$.

2. We develop computationally tractable gradient descent algorithms for WDPO and KLDPO that can be seamlessly integrated into existing LLM alignment pipelines.

3. Empirically, we show that standard DPO is sensitive to preference distribution shift, leading to degraded performance when training and evaluation rewards differ. In contrast, our robust variants, WDPO and KLDPO, consistently achieve superior performance across diverse alignment tasks. For example, we fine-tune LLaMA-3.2-1B/3B-Instruct and LLaMA-3.1-8B-Instruct models on prompts from the HelpSteer2 dataset (Wang et al., 2024b) using preferences generated by the ArmoRM reward model (Wang et al., 2024a), and evaluate them on distinct reward objectives from the OpenLLM Leaderboard (Fourrier et al., 2024).

## 2   Related Work

**Robust RLHF:** Bai et al. (2022) proposed to adjust weights on the combination of loss functions based on different topics (harmless vs. helpful) for robust reward learning. Chakraborty et al. (2024) proposed to learn multiple reward functions for different sub-populations through an expectation-maximization approach, and a robust policy based on these rewards via a max-min optimization, which is different from our distributional robust learning approach. Padmakumar et al. (2024) augmented the existing binary preference datasets with synthetic preference judgments to estimate the diversity of user preferences. Yan et al. (2024) proposed a Bayesian reward model ensemble to quantify the uncertainty of reward estimation and used it to reduce reward overoptimization. Bukharin et al. (2024) proposed a robust RLHF approach for addressing the preference data corruption problem.

**Robust DPO:** Huang et al. (2025) proposed $\chi$PO that implements the principle of pessimism in the face of uncertainty via regularization with the $\chi^2$-divergence for avoiding reward hacking/overoptimization w.r.t. the estimated reward. Ramesh et al. (2024) proposed Group Robust Preference Optimization (GRPO) to address the diverse preference problem by modeling the total loss as a weighted sum of individual DPO losses computed on separate preference datasets, and optimizing for the worst-case weighting. In contrast, our approach does not assume access to such a group structure and instead directly models distributional robustness over a single dataset that implicitly aggregates diverse preferences. Chowdhury et al. (2024) considered the setting where $\epsilon$-fraction of the preference labels in the training dataset is corrupted and proposed a noise-robust algorithm to mitigate its effect, assuming the knowledge of $\epsilon$. Wu et al. (2024) focused on adapting the DPO penalty parameter $\beta$ to handle varying data quality within the training set. The most related work is Wu et al. (2025), which applies distributional robustness to mitigate data corruption and noise in preference data. Unlike our work, it does not address distribution shift or provide theoretical guarantees, and lacks empirical evaluation on preference distribution shift. Concurrent to our work, Mandal et al. (2025) proposed a distributionally robust version of RLHF and DPO, using total variation (TV) uncertainty sets. However, their theoretical analysis offers the natural policy gradient (NPG) style optimization convergence guarantees for the loss function. In contrast, we go one step further: by leveraging strong convexity, we establish finite-sample guarantees not just for the learning loss, but for convergence of the policy parameters. Our analysis is algorithm-agnostic and applies to any solver capable of optimizing the robust DPO loss. Additionally, our formulation uses KL and Wasserstein uncertainty sets, which are more standard in large-scale LLM alignment.

**Distributionally Robust Learning:** Distributionally robust learning is a statistical learning framework designed to enhance model performance under distributional shifts between training and test data (Chen and Paschalidis, 2018). It employs a minimax approach where an adversary maximizes the expected loss by shifting the test distribution within a specified uncertainty set, while the learner minimizes this adversarial loss. This approach using the $f$-divergence (Namkoong and Duchi, 2016; Duchi and Namkoong, 2021; Levy et al., 2020) and the Wasserstein metric (Mohajerin Esfahani

and Kuhn, 2018; Kuhn et al., 2019; Gao et al., 2022) have gained significant attention recently. Distributionally robust algorithms have been developed to address problems in supervised learning (Chen and Paschalidis, 2018; Namkoong and Duchi, 2016; Levy et al., 2020), imitation learning (Bashiri et al., 2021; Panaganti et al., 2023), multi-armed bandits (Si et al., 2020; Yang et al., 2023), and reinforcement learning (Panaganti et al., 2022; Zhou et al., 2024; Shi and Chi, 2024; Yang et al., 2022; Panaganti et al., 2025).

## 3  Preliminaries

**Notations:**  We use calligraphic letters for sets, e.g., $\mathcal{S}$. $\|\cdot\|$ denotes the Euclidean norm. When $\Sigma$ is a positive semi-definite matrix, we write $\|x\|_\Sigma = \sqrt{x^\top \Sigma x}$ as a semi-norm of $x$. For any measure P, we use $\mathsf{P}_n$ to denote the empirical distribution constructed using $n$ i.i.d. samples, $x_1, \ldots, x_n$, from P, i.e., $\mathsf{P}_n = (1/n)\sum_{i=1}^n \delta_{x_i}$, where $\delta_x$ is the Dirac measure. We use $\sigma$ to denote the sigmoid (standard logistic) function. We use $l(z;\theta)$ and $l_z(\theta)$ to denote the loss incurred by sample $z$ with policy parameter $\theta$. For any set $\mathcal{Z}$, $\mathcal{P}(\mathcal{Z})$ is the set of all Borel measures over $\mathcal{Z}$. For any positive semi-definite matrix $\Sigma$, $\lambda_{\min}(\Sigma)$ and $\lambda_{\max}(\Sigma)$ denote its smallest and largest eigenvalues.

**Wasserstein Distance:**  For a given set $\mathcal{Z}$, equipped with a metric $d$, the Wasserstein distance of order $p$ between two distributions $\mu, \nu \in \mathcal{P}(\mathcal{Z})$ is defined as (see Villani et al. (2009)):

$$\mathsf{W}_p(\mu, \nu) = \min_{\gamma \in \mathcal{P}(\mathcal{Z} \times \mathcal{Z})} \left\{ \int_{\mathcal{Z} \times \mathcal{Z}} d^p(x, x')\gamma(dx, dx') \colon \gamma \text{ has marginal distributions } \mu, \nu \right\}.$$

**Kullback-Leibler Divergence:**  For any two probability distributions P and Q defined on $\mathcal{Z}$, the Kullback-Leibker (KL) divergence is defined as $D_{\mathrm{KL}}(\mathsf{P} \parallel \mathsf{Q}) = \sum_{z \in \mathcal{Z}} \mathsf{P}(z) \log(\mathsf{P}(z)/\mathsf{Q}(z))$.

**Reinforcement Learning from Human Feedback:**  The RLHF paradigm consists of three steps:

*Step 1: Supervised Fine-tuning (SFT).* SFT involves fine-tuning a pre-trained LLM through supervised learning on high-quality data, curated for the downstream tasks.

*Step 2: Reward Modelling.* In the second step, given any context $s \in \mathcal{S}$, two responses $a^1, a^2 \in \mathcal{A}$ are independently sampled from the behavior policy $\pi^o$ (typically the SFT policy $\pi_{\mathrm{SFT}}$). Then, a (human) labeler provides a preference response between these responses. We assume that the preference responses are generated according to the Bradley-Terry (BT) model (Bradley and Terry, 1952):

$$P^*(a^1 \succ a^2 \mid s) = \frac{\exp\left(r^*(s, a^1)\right)}{\exp\left(r^*(s, a^1)\right) + \exp\left(r^*(s, a^2)\right)}, \tag{1}$$

where $a^1 \succ a^2$ denotes $a^1$ being preferred over $a^2$, and $r^*$ is the underlying unknown reward function. We use $a^w, a^l$ to denote the preferred and dis-preferred responses, respectively. We assume access to a static dataset of comparison, $\mathcal{D} = \{(s_i, a_i^w, a_i^l)\}_{i=1}^n$, where $s_i$'s are sampled from some initial prompt (context) distribution $\mu^o$, $a_i^1, a_i^2$'s are independently sampled from $\pi_{\mathrm{SFT}}$, and the preferences responses are sampled from the BT model $P^*$. With $\mathcal{D}$, we can learn a parameterized reward model $r_\phi(s, a)$ by minimizing the maximum likelihood estimation (MLE) loss,

$$\mathcal{L}^{\mathrm{RLHF}}(r_\phi; \mathcal{D}) = -\mathbb{E}_{(s, a^w, a^l) \sim \mathcal{D}}[\log \sigma(r_\phi(s, a^w) - r_\phi(s, a^l))].$$

*Step 3: RL Fine-Tuning.* In the final step, the optimal policy $\pi^*$ under the reward $r_\phi$ is obtained by solving the KL-regularized reward maximization problem given by

$$\max_\pi \mathbb{E}_{s \sim \mu} \left[ \mathbb{E}_{a \sim \pi(\cdot|s)}[r_\phi(s, a)] - \beta D_{\mathrm{KL}}(\pi(\cdot \mid s) \parallel \pi_{\mathrm{ref}}(\cdot \mid s)) \right], \tag{2}$$

where $\beta$ is a parameter controlling the deviation from the base reference policy $\pi_{\mathrm{ref}}$.

**Direct Preference Optimization (DPO):**  The DPO approach (Rafailov et al., 2023) leverages the fact that the unknown reward function can be expressed in terms of the optimal policy and the reference policy. Formally, given any reward function $r^*$, the optimal solution of Eq. (2) takes the form $\pi^*(a \mid s) = \frac{1}{Z^*(s)}\pi_{\mathrm{ref}}(a \mid s)\exp\left(r^*(s, a)/\beta\right)$, where $Z^*(s)$ denotes the partition (normalizing) function. Rearranging the above, we get $r^*(s, a) = \beta \log \frac{\pi^*(a|s)}{\pi_{\mathrm{ref}}(a|s)} + \beta \log Z^*(s)$ for all $(s, a)$. Substituting this into Eq. (1), the optimal RLHF policy $\pi^*$ satisfies the preference model:

$$P^*(a^1 \succ a^2 \mid s) = \sigma\left(\beta \log \frac{\pi^*(a^1 \mid s)}{\pi_{\mathrm{ref}}(a^1 \mid s)} - \beta \log \frac{\pi^*(a^2 \mid s)}{\pi_{\mathrm{ref}}(a^2 \mid s)}\right).$$

Using the preference response dataset $\mathcal{D}$, we can learn the optimal policy directly by minimizing the MLE loss for a parameterized policy $\pi_\theta$,

$$\mathcal{L}^{\mathrm{DPO}}(\pi_\theta; \mathcal{D}) = -\mathbb{E}_{(s,a^w,a^l)\sim\mathcal{D}}\left[\log\sigma\left(\beta\log\frac{\pi_\theta(a^w\mid s)}{\pi_{\mathrm{ref}}(a^w\mid s)} - \beta\log\frac{\pi_\theta(a^l\mid s)}{\pi_{\mathrm{ref}}(a^l\mid s)}\right)\right]. \tag{3}$$

**Distributional Uncertainty Sets:** Given any $\rho > 0$ and $\mathsf{P}^o \in \mathcal{P}(\mathcal{Z})$, we define the distributional uncertainty set as

$$\mathcal{P}(\rho; \mathsf{P}^o) := \{\mathsf{P} \in \mathcal{P}(\mathcal{Z})\colon D(\mathsf{P}, \mathsf{P}^o) \leq \rho\}, \tag{4}$$

where $D(\cdot, \cdot)$ is some distance metric between two probability measures, e.g., $\mathsf{W}_p$ and $D_{\mathrm{KL}}$.

# 4 Distributionally Robust DPO

In this section, we formulate our Wasserstein DPO (WDPO) and Kullback-Leibler DPO (KLDPO).

**Sampling Procedure:** As described in Section 3, a prompt $s \in \mathcal{S}$ is drawn from an initial distribution $\mu^o$, and two responses $a^1, a^2 \sim_{\mathrm{i.i.d.}} \pi^o(\cdot \mid s)$ are sampled independently (with $\pi^o = \pi_{\mathrm{SFT}}$ in practice). Following Zhu et al. (2023), we define $y \in \{0, 1\}$ to indicate preference: $y = 1$ if $a^1 \succ a^2 \mid s$ and $y = 0$ otherwise. The label $y$ is drawn from a Bernoulli distribution defined by the BT model $P^*$. The full data-generating distribution is given below.

**Definition 1** (Joint data-generating distribution). *Consider the product space $\mathcal{Z} := \mathcal{S} \times \mathcal{A} \times \mathcal{A} \times \{0, 1\}$. We define the nominal data-generating distribution as*

$$\mathsf{P}^o(s, a^1, a^2, y) = \mu^o(s)\pi^o(a^1 \mid s)\pi^o(a^2 \mid s) \cdot [\mathbb{1}_{\{y=1\}}P^*(a^1 \succ a^2 \mid s) + \mathbb{1}_{\{y=0\}}P^*(a^2 \succ a^1 \mid s)].$$

We will also denote $z = (s, a^1, a^2, y) \in \mathcal{Z}$ and $\mathsf{P}^o(z) = \mathsf{P}^o(s, a^1, a^2, y)$. We assume that $\mathsf{P}^o$ generates the dataset $\mathcal{D} = \{z_i\}_{i=1}^n$ used for learning, i.e., $z_i \sim \mathsf{P}^o$.

## 4.1 Distributionally Robust DPO

From the DPO objective (Eq. (3)), we define the *pointwise* DPO loss function as follows

$$l(z; \theta) = -y\log\sigma(\beta h_\theta(s, a^1, a^2)) - (1-y)\log\sigma(\beta h_\theta(s, a^2, a^1)), \tag{5}$$

where $h_\theta(s, a^1, a^2) := \log\frac{\pi_\theta(a^1|s)}{\pi_{\mathrm{ref}}(a^1|s)} - \log\frac{\pi_\theta(a^2|s)}{\pi_{\mathrm{ref}}(a^2|s)}$ is the *preference score* of an answer $a^1$ relative to another one $a^2$ (but parameterized in policy parameter $\theta$). Let $\mathcal{P}(\rho; \mathsf{P}^o)$ be a distributional uncertainty set centered around $\mathsf{P}^o$ with radius $\rho > 0$. Following the principles of distributionally robust optimization (DRO), we formulate the distributionally robust DPO objective as:

$$\min_\theta \max_{\mathsf{P}\in\mathcal{P}(\rho;\mathsf{P}^o)} \mathbb{E}_{z\sim\mathsf{P}}[l(z;\theta)]. \tag{6}$$

Intuitively, we aim to find the best policy under the worst-case data distribution.

When we have a Wasserstein uncertainty set $\mathcal{P}_{\mathsf{W}_p}$, i.e., Eq. (4) equipped with the $p$-th order Wasserstein distance, we define the Wasserstein DPO (WDPO) loss as follows

$$\mathcal{L}^{\mathrm{W}}(\theta; \rho) = \sup_{\mathsf{P}\in\mathcal{P}_{\mathsf{W}_p}(\rho;\mathsf{P}^o)} \mathbb{E}_{z\sim\mathsf{P}}[l(\theta; z)], \tag{7}$$

Similarly, given a Kullback-Leibler uncertainty set $\mathcal{P}_{\mathrm{KL}}(\rho; \mathsf{P}^o)$, we define the KLDPO loss as follows

$$\mathcal{L}^{\mathrm{KL}}(\theta; \rho) = \sup_{\mathsf{P}\in\mathcal{P}_{\mathrm{KL}}(\rho;\mathsf{P}^o)} \mathbb{E}_{z\sim\mathsf{P}}[l(\theta; z)]. \tag{8}$$

When the nominal distribution $\mathsf{P}^o$ is replaced with its empirical counterpart, i.e., $\mathsf{P}_n^o := (1/n)\sum_{i=1}^n \delta_{z_i}$, where $z_1, \ldots, z_n$ are $n$ i.i.d. samples from $\mathsf{P}^o$, we use $\mathcal{L}_n^{\mathrm{W}}(\theta; \rho)$ and $\mathcal{L}_n^{\mathrm{KL}}(\theta; \rho)$ to denote the empirical WDPO and KLDPO losses incurred by the policy parameter $\theta$, respectively.

# 5 Theoretical Analysis

In this section, we present the sample complexity guarantees for our WDPO and KLDPO algorithms. We make the following assumptions for the rest of the papers.

**Assumption 1** (Log-linear policy class). *Let $\psi \colon \mathcal{S} \times \mathcal{A} \to \mathbb{R}^d$ be a known $d$-dimensional feature mapping with $\max_{s,a}\|\psi(s,a)\|_2 \leq 1$. Assume a bounded policy parameter set $\Theta := \{\theta \in \mathbb{R}^d : \|\theta\|_2 \leq B\}$. We consider the following class of log-linear policies:*

$$\Pi = \left\{ \pi_\theta \colon \pi_\theta(a \mid s) = \frac{\exp\left(\theta^\top \psi(s,a)\right)}{\sum_{a' \in \mathcal{A}} \exp\left(\theta^\top \psi(s,a')\right)} \right\}. \tag{9}$$

**Remark 1.** *This is a standard assumption in the theoretical analysis of the RL algorithms (Agarwal et al., 2021; Modi et al., 2020), RLHF (Zhu et al., 2023), and DPO (Nika et al., 2024; Chowdhury et al., 2024). Our analysis can be extended to the neural policy class where $\theta^\top \psi(s,a)$ is replaced $f_\theta(s,a)$, where $f_\theta$ is a neural network with twice differentiability and smoothness assumptions.*

We also make the following data coverage assumption on the uncertainty set $\mathcal{P}(\rho; \mathsf{P}^o)$.

**Assumption 2** (Regularity condition). *There exists $\lambda > 0$ such that*

$$\Sigma_\mathsf{P} := \mathbb{E}_{(s,a^1,a^2,y)\sim\mathsf{P}}[(\psi(s,a^1) - \psi(s,a^2))(\psi(s,a^1) - \psi(s,a^2))^\top] \succeq \lambda I, \quad \forall \mathsf{P} \in \mathcal{P}(\rho; \mathsf{P}^o).$$

**Remark 2.** *We note that similar assumptions on data coverage under linear architecture models are standard in the offline RL literature (Agarwal et al., 2019; Wang et al., 2021; Jin et al., 2021). Implicitly, Assumption 2 imposes $\lambda \leq \lambda_{\min}(\Sigma_{\mathsf{P}^o})$, which means that the data-generating distribution $\mathsf{P}^o$ has good coverage.*

### 5.1 Estimation Error for WDPO

Let $\theta^* \in \arg\min_{\theta \in \Theta} \mathcal{L}^{\mathrm{DPO}}(\theta)$ be the ground-truth optimal policy parameter with respect to the true nominal distribution and let its empirical counterpart be $\theta_n \in \arg\min_{\theta \in \Theta} \mathcal{L}_n^{\mathrm{DPO}}(\theta)$. Now for the robust policy parameters, we let $\theta^{\mathrm{W}} \in \arg\min_{\theta \in \Theta} \mathcal{L}^{\mathrm{W}}(\theta; \rho)$, and let its empirical counterpart be $\theta_n^{\mathrm{W}} \in \arg\min_{\theta \in \Theta} \mathcal{L}_n^{\mathrm{W}}(\theta; \rho)$. Now, present our main result on the sample complexity result for the convergence of the robust policy parameter.

**Theorem 1** (Estimation error of $\theta_n^{\mathrm{W}}$). *Let $\delta \in (0,1)$. With probability at least $1 - \delta$, we have*

$$\|\theta_n^{\mathrm{W}} - \theta^{\mathrm{W}}\|_2^2 \leq \sqrt{\frac{8K^2 \log(2/\delta)}{\gamma^2 \lambda^2 n}},$$

*where $\gamma = \frac{\beta^2 e^{4\beta B}}{(1 + e^{4\beta B})^2}$ and $K = |\log \sigma(-4\beta B)|$, $\lambda$ is the regularity number defined in Assumption 2.*

*Proof sketch.* Strong duality of Wasserstein DRO (see Gao and Kleywegt (2022) and Corollary 1) helps us reduce the difference $\left|\mathcal{L}^{\mathrm{W}}(\theta; \rho) - \mathcal{L}_n^{\mathrm{W}}(\theta; \rho)\right|$ to the concentration $\left|\mathbb{E}_{z\sim\mathsf{P}^o}[l_\eta(z; \theta)] - \mathbb{E}_{z\sim\mathsf{P}_n^o}[l_\eta(z; \theta)]\right|$, where $l_\eta(z; \theta) = \inf_{z\in\mathcal{Z}}[\eta d^p(z, z') - l(z; \theta)]$ is called the *Moreau-Yosida regularization* of $-l$ with parameter $1/\eta$. We show that, for all $\eta \geq 0$, all $l_\eta$ are uniformly bounded. We then use Hoeffding's inequality to obtain concentration. Detailed proof is in Appendix B.2.

Next, when Assumption 2 is in place, we can show that $g(\theta) := \mathbb{E}_{z\sim\mathsf{P}}[l(z; \theta)]$ is $\gamma$-strongly convex w.r.t. the positive definite norm $\|\cdot\|_{\Sigma_\mathsf{P}}$. Further, by the property of supremum, we can show that $\mathcal{L}^{\mathrm{W}}$ is $\gamma\lambda$-strongly convex but w.r.t. $\|\cdot\|_2$. A detailed proof is provided in Appendix B.3.

Decompose $\mathcal{L}^{\mathrm{W}}(\theta_n^{\mathrm{W}}) - \mathcal{L}^{\mathrm{W}}(\theta^{\mathrm{W}})$ into three terms: $\mathcal{L}^{\mathrm{W}}(\theta_n^{\mathrm{W}}; \rho) - \mathcal{L}_n^{\mathrm{W}}(\theta_n^{\mathrm{W}}; \rho)$, $\mathcal{L}_n^{\mathrm{W}}(\theta_n^{\mathrm{W}}; \rho) - \mathcal{L}_n^{\mathrm{W}}(\theta^{\mathrm{W}}; \rho)$, and $\mathcal{L}_n^{\mathrm{W}}(\theta^{\mathrm{W}}; \rho) - \mathcal{L}^{\mathrm{W}}(\theta^{\mathrm{W}}; \rho)$. The second term is non-positive since $\theta_n^{\mathrm{W}}$ is the minimizer of $\mathcal{L}_n^{\mathrm{W}}$. Now we apply the concentration of the WDPO loss function (see Lemma 9 in Appendix B.2) to $|\mathcal{L}^{\mathrm{W}}(\theta_n^{\mathrm{W}}; \rho) - \mathcal{L}_n^{\mathrm{W}}(\theta_n^{\mathrm{W}}; \rho)|$ and $|\mathcal{L}_n^{\mathrm{W}}(\theta^{\mathrm{W}}; \rho) - \mathcal{L}^{\mathrm{W}}(\theta^{\mathrm{W}}; \rho)|$. Finally, we use the property of strongly convex function (Lemma 5) on $\mathcal{L}^{\mathrm{W}}$ to acquire the policy parameter convergence. The detailed proof is in Appendix B.4. $\qquad\square$

We state the convergence result for DPO to facilitate comparison with its robust counterpart.

**Proposition 1** (Estimation error of (non-robust) DPO). *Let $\delta \in (0,1)$ and $\beta > 0$.*

$$\|\theta_n - \theta^*\|_{\Sigma_\mathcal{D} + \lambda I} \leq 2\sqrt{\frac{4\beta^2}{\gamma^2 n}(d + \log(1/\delta)) + 2\lambda B^2},$$

*with probability at least $1 - \delta$ and where $\gamma = \frac{\beta^2 e^{4\beta B}}{(1 + e^{4\beta B})^2}$, and $\Sigma_\mathcal{D} = \frac{1}{n}\sum_{i=1}^n (\psi(s_i, a_i^1) - \psi(s_i, a_i^2))(\psi(s_i, a_i^1) - \psi(s_i, a_i^2))^\top$ is the sample covariance matrix.*

| **Algorithm 1** WDPO Algorithm | **Algorithm 2** KLDPO Algorithm |
|---|---|
| 1: **Input:** Dataset $\mathcal{D} = \{(s_i, a_i^w, a_i^l)\}_{i=1}^n$, reference policy $\pi_{\text{ref}}$, robustness hyperparameter $\rho_o$, learning rate $\eta$, initial policy $\pi_\theta$. | 1: **Input:** Dataset $\mathcal{D} = \{(s_i, a_i^w, a_i^l)\}_{i=1}^n$, reference policy $\pi_{\text{ref}}$, robustness temperature parameter $\tau$, learning rate $\eta$, initial policy $\pi_\theta$. |
| 2: **while** $\theta$ has not converged **do** | 2: **while** $\theta$ has not converged **do** |
| 3:    Calculate the non-robust DPO loss $\mathcal{L}^{\text{DPO}}(\pi_\theta; \mathcal{D})$ according to Eq. (3) | 3:    Approximate the worst-case kernel |
| 4:    Calculate the gradient regularizer loss $$\mathcal{R}(\pi_\theta; \mathcal{D}) = \rho_o(\mathbb{E}_{z\sim\mathcal{D}}\|\nabla_z l(z;\theta)\|_2^2)^{1/2}$$ | $$\underline{\mathsf{P}}(i) \propto \exp((1/\tau)(l(z_i;\theta) - (1/n)\textstyle\sum_{i=1}^n l(z_i;\theta)))$$ |
| 5:    Calculate the approximate WDPO loss $$\mathcal{L}^{\text{W}}(\theta;\mathcal{D}) \coloneqq \mathcal{L}^{\text{DPO}}(\pi_\theta;\mathcal{D}) + \mathcal{R}(\pi_\theta;\mathcal{D})$$ | 4:    Calculate the approximate KLDPO loss $$\mathcal{L}^{\text{KL}}(\theta;\mathcal{D}) \coloneqq \textstyle\sum_{i=1}^n \underline{\mathsf{P}}(i) \cdot l(z_i;\theta)$$ |
| 6:    $\theta \leftarrow \theta - \eta\nabla_\theta\mathcal{L}^{\text{W}}(\theta;\mathcal{D})$ | 5:    $\theta \leftarrow \theta - \eta\nabla_\theta\mathcal{L}^{\text{KL}}(\theta;\mathcal{D})$ |
| 7: **Output:** $\pi_\theta$ | 6: **Output:** $\pi_\theta$ |

A matching result can be derived as a special case of Chowdhury et al. (2024, Theorem 4.2). We provide an independent proof with precise constants in Appendix B.1.

**Remark 3.** *We would like to note that the estimation error rate of convergence for WDPO is $\|\theta_n^{\text{W}} - \theta^{\text{W}}\|_2 = O(n^{-1/4})$, from Theorem 1. The estimation error rate of convergence for (non-robust) DPO is $\|\theta_n - \theta^*\|_{\Sigma_{\mathcal{D}}+\lambda I} = O(n^{-1/2})$, from Proposition 1. So, the estimation error rate of convergence for WDPO is worse than that of (non-robust) DPO. This arises due to significant challenges exclusive to the robust setting. For example, for the non-robust DPO, we can calculate the closed-form expression of $\nabla_\theta(1/n)\sum_{i=1}^n l(z_i;\theta)$ (see Eq. (20)). This allows us to write $\|\nabla_\theta(1/n)\sum_{i=1}^n l(z_i;\theta^*)\|_{(\Sigma_{\mathcal{D}}+\lambda I)^{-1}}$ in quadratic form and then obtain a concentration using Bernstein's inequality. However, for WDPO, we note that $\nabla_\theta\mathcal{L}_n^{\text{W}}(\theta^{\text{W}}) \neq \sup_{\mathsf{P}\in\mathcal{P}_{\mathsf{W}_p}}\nabla_\theta\mathbb{E}_{z\sim\mathsf{P}}[l(z;\theta^{\text{W}})]$, and the non-robust approach will not work for the robust setting. Developing analysis techniques to achieve a better rate of convergence for robust DPO is an open question.*

### 5.2 Estimation Error for KLDPO

Let $\theta^{\text{KL}} \in \arg\min_{\theta\in\Theta}\mathcal{L}^{\text{KL}}(\theta;\rho)$, and let its empirical counterpart be $\theta_n^{\text{KL}} \in \arg\min_{\theta\in\Theta}\mathcal{L}_n^{\text{KL}}(\theta;\rho)$. The convergence analysis for the KLDPO loss and policy parameter closely parallels that of Wasserstein DPO. We present the main theorems below and defer detailed proofs to Appendix C.

**Theorem 2** (Estimation error of $\theta_n^{\text{KL}}$). *Let $\delta \in (0,1)$. With probability at least $1 - \delta$, we have*

$$\|\theta_n^{\text{KL}} - \theta^{\text{KL}}\|_2^2 \leq \sqrt{\frac{8\overline{\lambda}^2\exp(L/\underline{\lambda})\log(2/\delta)}{\gamma^2\lambda^2 n}},$$

*where $\gamma = \frac{\beta^2 e^{4\beta B}}{(1+e^{4\beta B})^2}$. $\lambda$ is the regularity condition number defined in Assumption 2, $0 < \lambda \leq \lambda_{\min}(\Sigma_{\mathsf{P}^o})$. $\underline{\lambda}, \overline{\lambda}$ are some universal constants, and $L$ is an upper bound on the loss function $l$.*

**Remark 4.** *The exponential constant in the upper bound is a characteristic of distributional robust optimization with KL uncertainty set Hu and Hong (2013, Proposition 2). Similar exponential constants appear in the theoretical analysis of the distributionally robust RL (Zhou et al., 2021; Yang et al., 2022; Panaganti and Kalathil, 2022; Xu et al., 2023). Both WDPO and KLDPO have $O(n^{-1/4})$ policy parameter convergence. An empirical comparison is given in Section 7.*

## 6 Tractable (Approximate) Algorithms

While our distributionally robust DPO formulations enjoy finite-sample guarantees, it is computationally challenging to solve the min-max objective of Eq. (6) using stochastic gradient descent methods. Though many min-max optimization problems can be solved by alternating gradient descent methods, our problem is not directly amenable to such an approach as we do not have direct control over the data distribution $\mathsf{P} \in \mathcal{P}(\rho;\mathsf{P}^o)$ which is not parameterized. Moreover, the preference data are generated according to the nominal distribution $\mathsf{P}^o$, and we do not have data samples from any other

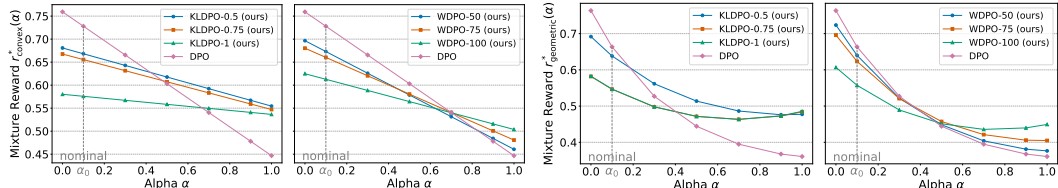

Figure 2: *DPO, WDPO, and KLDPO in Emotion Alignment.* Models are trained on preferences derived from convex (left two plots) and geometric (right two plots) mixtures of *anger* and *fear* objectives from the Emotion dataset (Saravia et al., 2018). To simulate preference shift, evaluation is performed at mixing coefficients $\alpha \neq \alpha_o$, where $\alpha_o = 0.1$ is used during training. We evaluate WDPO with robustness parameter $\rho_o \in \{50, 75, 100\}$ and KLDPO with robustness temperature $\tau \in \{0.5, 0.75, 1\}$. Additional experimental details are provided in Section 7.1.

distributions in the uncertainty set $\mathcal{P}(\rho; \mathsf{P}^o)$. To overcome this challenge, we introduce principled tractable algorithms to solve WDPO and KLDPO.

**Tractable WDPO:** The connection between Wasserstein distributionally robust optimization (DRO) and regularization has been established in various settings by many (Mohajerin Esfahani and Kuhn, 2018; Shafieezadeh-Abadeh et al., 2019; Chen and Paschalidis, 2018). We leverage the recent progress in Wasserstein theory on connecting Wasserstein distributionally robust optimization to regularization. For $p$-Wasserstein DRO, $p \in (1, \infty]$, Gao et al. (2022) shows that for a broad class of loss functions, possibly non-convex and non-smooth, with high probability, the Wasserstein DRO is asymptotically equivalent to variation regularization. In particular, an immediate consequence of Gao et al. (2022, Theorem 1) is that, when $p = 2$,

$$\min_{\theta \in \Theta} \sup_{\mathsf{P}:\, W_p(\mathsf{P}, \mathsf{P}_n^o) \leq \rho_n} \mathbb{E}_{z \sim \mathsf{P}}[l(z; \theta)] = \min_{\theta \in \Theta} \left\{ \mathbb{E}_{z \sim \mathsf{P}_n^o}[l(z; \theta)] + \rho_n \sqrt{(1/n) \sum_{i=1}^n \|\nabla_z l(z_i; \theta)\|_2^2} \right\} + O_p(1/n),$$

where $\rho_n = O(1/\sqrt{n})$. That is, one can solve the Wasserstein DRO objective by adding a gradient regularization to the empirical risk minimization (ERM) loss, $\mathbb{E}_{z \sim \mathsf{P}_n^o}[l(z; \theta)]$. Based on this, we propose a tractable WDPO algorithm in Algorithm 1. Note that the gradient regularizer has a sample-size-dependent coefficient. In practice, we absorb the factor $\rho_n/\sqrt{n}$ into $\rho_o$, which we treat as a tunable robustness hyperparameter.

**Tractable KLDPO:** The following proposition shows that we can approximate the worst-case probability distribution in a KL uncertainty set w.r.t. a given loss function. Similar results can also be found in distributionally robust reinforcement learning literature (e.g., Gadot et al. (2024)).

**Proposition 2** (Worst-case distribution (informal)). *Let $\underline{\mathsf{P}} \in \mathbb{R}^n$ be the worst-case distribution w.r.t. a loss function $l$ and KL uncertainty around the empirical distribution $\mathsf{P}_n^o$, defined as $\underline{\mathsf{P}} = \sup_{\mathsf{P}:\, D_{\mathrm{KL}}(\mathsf{P} \| \mathsf{P}_n^o) \leq \rho} \mathbb{E}_{z \sim \mathsf{P}}[l(z; \theta)]$. The worst-case distribution $\underline{\mathsf{P}}$ is related to $\mathsf{P}_n^o$ through*

$$\underline{\mathsf{P}}(i) \propto \mathsf{P}_n^o(i) \cdot \exp((1/\tau)(l(z_i; \theta) - \textstyle\sum_{i=1}^n \mathsf{P}_n^o(i) l(z_i; \theta))),$$

*where $\tau > 0$ is some constant.*

We defer the formal proof of Proposition 2 to Appendix D. It can be viewed as a re-weighting threshold: extreme losses are more biased towards the baseline empirical DPO loss. $\tau$ controls the intensity of re-weighting, acting as a temperature parameter. Based on Proposition 2, we propose a tractable KLDPO algorithm in Algorithm 2.

# 7 Experiments

We conduct experiments across three distinct settings that vary in dataset scale, model size, and the degree of distribution shift. For example, we fine-tune LLaMA-3.2-1B-Instruct, LLaMA-3.2-3B-Instruct, and LLaMA-3.1-8B-Instruct models on prompts from the HelpSteer2 dataset (Wang et al., 2024b), using preferences derived from the ArmoRM reward model (Wang et al., 2024a), and evaluate them on the OpenLLM Leaderboard v2 (Fourrier et al., 2024). Additional evaluations are provided in Appendix E. We provide the code at `https://github.com/TheBlackCat22/distributionally_robust_dpo`.

## 7.1 Experimental Setup

**Emotion Alignment:** We use the Emotion dataset (Saravia et al., 2018) to train a GPT-2 model (Radford et al., 2019) with a classification head for multi-label classification over five emotions: *sadness, joy, love, anger, fear*. The resulting sigmoid outputs are used as a multi-objective reward model for the remainder of this experiment. We also take a GPT-2 model and perform supervised fine-tuning (SFT) with the Emotion dataset to obtain our base model for preference alignment. To construct preference data, we mix objectives derived from our reward model. Specifically, we consider two reward objectives, $r_1, r_2$ and define two mixture reward functions (1) convex mixing $r^*_{\text{convex}}(\alpha) \coloneqq \alpha \cdot r_1 + (1 - \alpha) \cdot r_2$ and (2) geometric mixing $r^*_{\text{geometric}}(\alpha) \coloneqq r_1^{\alpha} \cdot r_2^{1-\alpha}$. For both reward functions, we generate two completions per prompt and assign preference labels using the Bradley-Terry (BT) model parameterized by $r^*(\alpha^o)$ for a chosen $\alpha^o \in [0, 1]$.

**ArmoRM Multi-objective Alignment:** We use the Absolute-Rating Multi-Objective Reward Model (ArmoRM) (Wang et al., 2024a) to define reward preferences, selecting pairs of equally weighted objectives (e.g., honesty, verbosity, safety) from its 19-dimensional first-stage outputs. Using Meta LLaMA-3.2-1B-Instruct as the base model, we generate two completions per prompt from the HelpSteer2 dataset (Wang et al., 2024b) and train models on preferences derived from the convex mixing of these reward pairs. We evaluate all models on five individual ArmoRM objectives, three of which are unseen during training, to simulate preference shift.

**Leaderboard Alignment:** We fine-tune LLaMA-3.2-1B-Instruct, LLaMA-3.2-3B-Instruct, and LLaMA-3.1-8B-Instruct models using preference data derived from the scalar rewards produced by the second stage of the ArmoRM reward model (Wang et al., 2024a). For each prompt from the HelpSteer2 dataset, we generate 10 responses, score them with ArmoRM, and constructe preference pairs by selecting the highest- and lowest-scoring completions. The models are evaluated on the OpenLLM Leaderboard v2 (Fourrier et al., 2024) using the LM Evaluation Harness (Gao et al., 2024).

## 7.2 Experiment Results

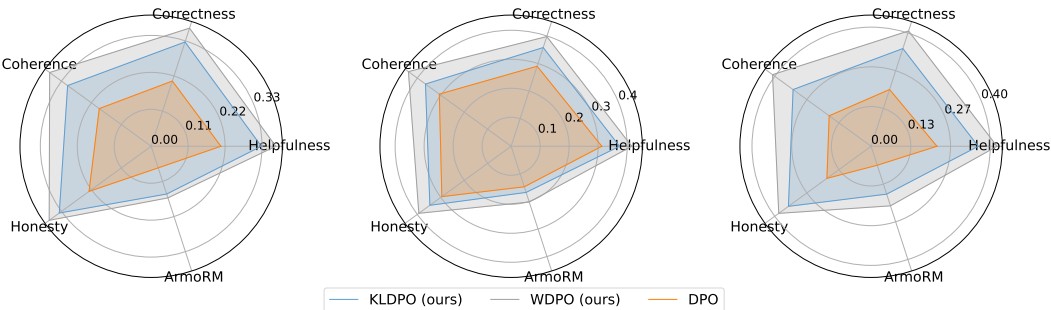

Figure 3: *DPO, WDPO, and KLDPO in ArmoRM Multi-objective Alignment.* LLaMA-3.2-1B-Instruct models are trained on preferences derived from three equally weighted objective pairs: (1) *Ultrafeedback-Truthfulness* and *Helpsteer-Complexity*, (2) *Ultrafeedback-Helpfulness* and *Helpsteer-Coherence*, and (3) *Helpsteer-Correctness* and *Helpsteer-Helpfulness* (left to right plots). We train all models for 4 epochs. To simulate preference shift, models are evaluated on five individual objectives, *Helpsteer-Helpfulness*, *Helpsteer-Correctness*, *Helpsteer-Coherence*, *Ultrafeedback-Honesty*, and the overall *ArmoRM* score, three of which were not used during training.

**Emotion Alignment Results:** In Fig. 2, we evaluate DPO, WDPO, and KLDPO under preference shifts between training and evaluation. All models are trained on preference labels emphasizing the emotion *fear*, while evaluation preferences gradually shift toward *anger*. The left two plots correspond to convex mixing of these emotions, and the right two use geometric mixing. As expected, DPO performs best when the evaluation preference closely matches the training setup. However, as the evaluation shifts toward *anger*, DPO's performance degrades significantly. In contrast, both WDPO and KLDPO maintain stable performance across the full range of evaluation preferences, consistently outperforming DPO under shift, demonstrating their robustness to preference misalignment.

**ArmoRM Multi-objective Alignment Results:** In Fig. 3, each radar plot corresponds to a different training reward pair, (1), (2), and (3), as defined in the figure caption. We evaluate all models on five individual ArmoRM objectives, three of which are unseen during training, to simulate preference shift.

| LLaMA-3.2-1B | IFEval | BBH | MATH | GPQA | MUSR | MMLU | LLaMA-3.2-3B | IFEval | BBH | MATH | GPQA | MUSR | MMLU |
|---|---|---|---|---|---|---|---|---|---|---|---|---|---|
| DPO at Epoch 2 (early stopping) | 0.48 | 0.35 | 0.08 | 0.27 | 0.35 | 0.17 | DPO | 0.55 | 0.45 | 0.08 | 0.24 | 0.36 | 0.30 |
| DPO at Epoch 4 (goodfit) | 0.48 | 0.34 | 0.07 | 0.26 | 0.33 | 0.17 | KLDPO ($\tau = 0.005$) | 0.74 | 0.46 | 0.19 | 0.26 | 0.35 | 0.32 |
| DPO at Epoch 6 (overfit) | 0.48 | 0.33 | 0.06 | 0.26 | 0.33 | 0.17 | WDPO ($\rho_o = 0.005$) | 0.62 | 0.45 | 0.06 | 0.25 | 0.36 | 0.30 |
| KLDPO ($\tau = 0.1$) | 0.53 | 0.36 | 0.08 | 0.25 | 0.33 | 0.18 | **LLaMA-3.1-8B** | **IFEval** | **BBH** | **MATH** | **GPQA** | **MUSR** | **MMLU** |
| KLDPO ($\tau = 0.05$) | 0.56 | 0.36 | 0.08 | 0.26 | 0.32 | 0.18 | DPO | 0.62 | 0.50 | 0.03 | 0.29 | 0.44 | 0.33 |
| WDPO ($\rho_o = 0.01$) | 0.52 | 0.36 | 0.09 | 0.25 | 0.34 | 0.19 | KLDPO ($\tau = 0.005$) | 0.72 | 0.51 | 0.24 | 0.29 | 0.34 | 0.37 |
| WDPO ($\rho_o = 0.005$) | 0.49 | 0.35 | 0.09 | 0.25 | 0.33 | 0.19 | KLDPO ($\tau = 0.01$) | 0.75 | 0.51 | 0.22 | 0.31 | 0.36 | 0.37 |

Table 1: *Evaluation of DPO, KLDPO, and WDPO on OpenLLM Leaderboard v2.* LLaMA-3.2-1B/3B-Instruct and LLaMA-3.1-8B-Instruct models are trained on preferences generated according to ArmoRM score and then evaluated on OpenLLM Leaderboard v2, which benchmarks LLMs across **six** tasks: *Massive Multitask Language Understanding* (MMLU), *Google-Proof Q&A Benchmark* (GPQA), *Multistep Soft Reasoning* (MUSR), *Mathematics Aptitude Test of Heuristics* (MATH), *Instruction Following Evaluation* (IFEval), and *Big Bench Hard* (BBH).

Across all settings, both KLDPO and WDPO consistently outperform DPO on all five evaluation axes, including those based on unseen objectives. This demonstrates their strong generalization and robustness to reward distribution shift, even when the evaluation preferences differ significantly from the training signal. Additional results are provided in Appendix E.1.

**Leaderboard Alignment Results:** Table 1 presents the performance of DPO, KLDPO, and WDPO on the OpenLLM leaderboard v2 (Fourrier et al., 2024). WDPO and KLDPO are trained for 2 epochs, matching DPO's optimal early-stopping point, which is a regularization technique to prevent overfitting. For LLaMA-3B and LLaMA-8B models, we align training durations similarly. Due to computational constraints, only KLDPO results are reported for the 8B model, given its scalability. These results, averaged over 39 subtasks, are supplemented by detailed evaluations in Appendix E.2, where WDPO and KLDPO demonstrate clear advantages across various subtasks.

# 8 Conclusions

We introduced a distributionally robust DPO framework, developed two scalable algorithms with theoretical guarantees, and integrated them into existing LLM alignment pipelines. Empirical results demonstrate their effectiveness under preference distribution shift. Future work includes extending our methods to mitigate reward hacking and generalizing robustness to other RLHF approaches.

# 9 Acknowledgments

The authors would like to thank Vishnu Teja Kunde for invaluable discussions. This work was supported in part by the National Science Foundation (NSF) grants NSF-CAREER-EPCN-2045783, ECCS-529620-00002, and CNS-526050-00002. Portions of this research were conducted with the advanced computing resources provided by Texas A&M High Performance Research Computing.

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

# A  Useful Technical Results

## A.1  Wasserstein Theory

We rely on the following strong duality result from the Wasserstein distributionally robust optimization (WDRO) literature.

**Lemma 1** (Gao and Kleywegt, 2022, Theorem 1; Strong Duality for DRO with Wasserstein Distance). *Consider any $p \in [1, \infty)$, any $\nu \in \mathcal{P}(\Xi)$, any $\rho > 0$, and any $\Psi \in L^1(\nu)$ such that the growth rate $\kappa$ of $\Psi$ satisfies*

$$\kappa := \inf \left\{ \eta \geq 0 \colon \int_\Xi \Phi(\eta, \zeta) \nu(d\zeta) > -\infty \right\} < \infty, \tag{10}$$

*where $\Phi(\eta, \zeta) := \inf_{\xi \in \Xi} \{ \eta d^p(\xi, \zeta) - \Psi(\xi) \}$ is a regularization operator. Then the strong duality holds with **finite optimal value** $v_p = v_D \leq \infty$, where*

$$v_p := \sup_{\mu \in \mathcal{P}(\Xi)} \left\{ \int_\Xi \Psi(\xi) \mu(d\xi) \colon \mathsf{W}_p(\mu, \nu) \leq \rho \right\}, \qquad \text{(Primal)}$$

$$v_D := \inf_{\eta \geq 0} \left\{ \eta \rho^p - \int_\Xi \inf_{\xi \in \Xi} [\eta d^p(\xi, \zeta) - \Psi(\xi)] \nu(d\zeta) \right\}. \qquad \text{(Dual)}$$

**Lemma 2** (Gao and Kleywegt, 2022, Lemma 2.(ii); Properties of the growth $\kappa$). *Suppose that $\nu \in \mathcal{P}_p(\Xi)$. Then the growth rate $\kappa$ (as defined in Eq. (10)) is finite if and only if there exists $\zeta^o \in \Xi$ and $L, M > 0$ such that*

$$\Psi(\xi) - \Psi(\zeta^o) \leq L d^p(\xi, \zeta^o) + M, \quad \forall \xi \in \Xi. \tag{11}$$

**Corollary 1.** *Consider any bounded loss function $l$ over bounded $\Xi$. Then the duality defined in Lemma 1 holds.*

*Proof.* It follows from Lemma 2. We can pick $L$ to be the diameter of $\Xi$ and $M$ to be the bound of $\Psi$. $\qquad\square$

## A.2  Optimization

**Lemma 3** (Beck, 2014, Theorem 1.24; Linear Approximation Theorem). *Let $f \colon U \to \mathbb{R}$ be a twice continuously differentiable function over an open set $U \subseteq \mathbb{R}^n$, and let $x, y \in U$ be such that $[x, y] \subseteq U$. Then there exists $\xi \in [x, y]$ such that*

$$f(y) = f(x) + \nabla f(x)^\top (y - x) + \frac{1}{2}(y - x)^\top \nabla^2 f(\xi)(y - x).$$

**Lemma 4** (Beck, 2017, Theorem 5.24; First-order characterizations of strong convexity). *Let $f \colon \mathbb{E} \to (-\infty, \infty]$ be a proper closed and convex function. Then for a given $\sigma > 0$, the following two claims are equivalent:*

*I. For any $x, y \in \mathbf{dom}(f)$ and $\lambda \in [0, 1]$:*

$$f(\lambda x + (1 - \lambda)y) \leq \lambda f(x) + (1 - \lambda)f(y) - \frac{\sigma}{2}\lambda(1 - \lambda)\|x - y\|^2.$$

*II.*

$$f(y) \geq f(x) + \langle g, y - x \rangle + \frac{\sigma}{2}\|y - x\|^2,$$

*for any $x \in \mathbf{dom}(\partial f)$, $y \in \mathbf{dom}(f)$ and $g \in \partial f(x)$.*

**Lemma 5** (Beck, 2017, Theorem 5.25; Existence and uniqueness of a minimizer of closed strongly convex functions). *Let $f \colon \mathbb{E} \to (-\infty, \infty]$ be a proper closed and $\sigma$-strongly convex function $\sigma > 0$. Then*

*I. $f$ has a unique minimizer;*

*II. $f(x) - f(x^*) \geq \frac{\sigma}{2}\|x - x^*\|^2$ for all $x \in \mathbf{dom}(f)$, where $x^*$ is the unique minimizer of $f$.*

### A.3 Distributionally Robust Optimization Results

The Kullback-Liebler uncertainty set can be constructed with the $f$-divergence. The $f$-divergence between the distribution $\mathsf{P}$ and $\mathsf{P}^o$ is defined as

$$D_f(\mathsf{P} \parallel \mathsf{P}^o) = \int_{\mathcal{X}} f\left(\frac{d\mathsf{P}}{d\mathsf{P}^o}\right) d\mathsf{P}^o, \tag{12}$$

where $f$ is a convex function. $f(t) = t\log(t)$ gives us the Kullback-Liebler divergence. Let $\mathsf{P}^o$ be a distribution on the space $\mathcal{X}$ and let $l\colon \mathcal{X} \to \mathbb{R}$ be a loss function. We have the following result from the distributionally robust optimization literature.

**Lemma 6** (Duchi and Namkoong, 2021, Proposition 1). *Let $D_f$ be the $f$-divergence defined in Eq. (12). Then,*

$$\sup_{\mathsf{P}\colon D_f(\mathsf{P} \parallel \mathsf{P}^o) \leq \rho} E_\mathsf{P}[l(X)] = \inf_{\lambda \geq 0, \eta \in \mathbb{R}} \mathbb{E}_{\mathsf{P}^o}\left[\lambda f^*\left(\frac{l(X) - \eta}{\lambda}\right)\right] + \lambda \rho + \eta, \tag{13}$$

*where $f^*(s) = \sup_{t \geq 0}\{st - f(t)\}$ is the Fenchel conjugate.*

### A.4 Concentration Results

**Lemma 7** (Hoeffding's inequality (see Boucheron et al., 2013, Theorem 2.8)). *Let $X_1, \ldots, X_n$ be independent random variables such that $X_i$ takes its values in $[a_i, b_i]$ almost surely for all $i \leq n$. Let*

$$S = \sum_{i=1}^n (X_i - \mathbb{E}\left[X_i\right]).$$

*Then for every $t > 0$,*

$$\mathbb{P}\left(S \geq t\right) \leq \exp\left(-\frac{2t^2}{\sum_{i=1}^n (b_i - a_i)^2}\right).$$

*Furthermore, if $X_1, \ldots, X_n$ are a sequence of independent, identically distributed random variables with mean $\mu$. Let $\overline{X}_n = \frac{1}{n}\sum_{i=1}^n X_i$. Suppose that $X_i \in [a, b], \forall i$. Then for all $t > 0$*

$$\mathbb{P}\left(\left|\overline{X}_n - \mu\right| \geq t\right) \leq 2\exp\left(-\frac{2nt^2}{(b-a)^2}\right).$$

**Lemma 8** (Hsu et al., 2012, Theorem 2.1). *Let $A \in \mathbb{R}^{n \times n}$ be a matrix, and let $\Sigma := A^\top A$. Suppose that $x = (x_1, \ldots, x_n)$ is a random vector such that for some $\mu \in \mathbb{R}^n$ and $\sigma \geq 0$,*

$$\mathbb{E}[\exp(\alpha^\top (x - \mu))] \leq \exp(\|\alpha\|^2 \sigma^2/2),$$

*for all $\alpha \in \mathbb{R}^n$. For all $t > 0$,*

$$\mathbb{P}\left[\|Ax\|^2 > \sigma^2 \cdot \left(\mathsf{Tr}(\Sigma) + 2\sqrt{\mathsf{Tr}(\Sigma^2)t} + 2\|\Sigma\|t\right) + \mathsf{Tr}(\Sigma\mu\mu^\top) \cdot \left(1 + 2\sqrt{\frac{t\|\Sigma\|^2}{\mathsf{Tr}(\Sigma^2)}}\right)\right] \leq e^{-t}.$$

*Moreover, if $\mu = 0$ and $\sigma = 1$, then the probability inequality reads*

$$\mathbb{P}\left(\|Ax\|^2 > \mathsf{Tr}(\Sigma) + 2\sqrt{\mathsf{Tr}(\Sigma^2)t} + 2\|\Sigma\|t\right) \leq e^{-t}.$$

## B Proof of WDPO Sample Complexity

Many properties of distributionally robust DPO are derived from those of the non-robust DPO. We hence start with the following proof of policy parameter convergence in the non-robust setting (Proposition 1).

### B.1 Proof of Non-robust DPO Policy Parameter Convergence

Recall the pointwise DPO loss:

$$l(\theta; s, a^1, a^2, y) := -y\log\sigma(\beta h_\theta(s, a^1, a^2)) - (1 - y)\log\sigma(\beta h_\theta(s, a^2, a^1)),$$

where $h_\theta(s, a^1, a^2) := \log\frac{\pi_\theta(a^1|s)}{\pi_{\text{ref}}(a^1|s)} - \log\frac{\pi_\theta(a^2|s)}{\pi_{\text{ref}}(a^2|s)}$. Denote this loss by $l_z(\theta)$ where $z = (s, a^1, a^2, y)$. We also denote the empirical (sample) DPO loss as

$$l_\mathcal{D}(\theta) = \frac{1}{n}\sum_{i=1}^n l_{z_i}(\theta) = \frac{1}{n}\sum_{i=1}^n -y_i\log\sigma(\beta h_\theta(s_i, a_i^1, a_i^2)) - (1 - y_i)\log\sigma(\beta h_\theta(s_i, a_i^2, a_i^1)).$$

We denote the MLE solution to $l_\mathcal{D}$ by $\theta_n^{\text{dpo}} \in \operatorname{argmin}_{\theta \in \Theta} l_\mathcal{D}(\theta)$. Also, denote the true parameter which is the global minimum of the population negative log likelihood by $\theta^*$.

**(Almost) Strong Convexity of $l$.** In order to calculate the Hessian matrix of $l_z$ w.r.t. $\theta$, we need to calculate $\nabla^2_\theta \log \sigma(\beta h_\theta(s, a^1, a^2))$.

Suppose $f\colon \mathbb{R} \to \mathbb{R}$, $g\colon \mathbb{R}^d \to \mathbb{R}$. The Hessian of $f \circ g$ is, for any $x \in \mathbb{R}^d$,

$$\nabla^2_x (f \circ g)(x) = f'(g(x))\nabla^2_x g(x) + f''(g(x))\nabla_x g(x)\nabla_x g(x)^\top. \tag{14}$$

Recall that $\sigma$ is the sigmoid function. It has the properties: $\sigma(-x) = 1 - \sigma(x)$ and $\sigma'(x) = \sigma(x)(1 - \sigma(x))$. Let $f(x) = \log \sigma(x)$, we have

$$\frac{d}{dx} f(x) = \frac{\sigma'(x)}{\sigma(x)} = \frac{\sigma(x)(1 - \sigma(x))}{\sigma(x)} = \sigma(-x);$$

$$\frac{d^2}{dx^2} f(x) = \frac{d}{dx}[\sigma(-x)] = \frac{d}{dx}[1 - \sigma(x)] = -\sigma'(x) = -\sigma(x)\sigma(-x).$$

With $g(\theta) := \beta h_\theta(s, a^1, a^2)$ and the Hessian chain rule for composition with a scalar function (Eq. (14)), we have

$$\nabla^2_\theta \log \sigma(\beta h_\theta(s, a^1, a^2)) = \beta\sigma(-\beta h_\theta(s, a^1, a^2))\nabla^2_\theta h_\theta(s, a^1, a^2)$$
$$- \beta^2\sigma(\beta h_\theta(s, a^1, a^2))\sigma(-\beta h_\theta(s, a^1, a^2))\nabla_\theta h_\theta(s, a^1, a^2)\nabla_\theta h_\theta(s, a^1, a^2)^\top.$$

In addition, we have the following observations

$$\nabla_\theta h_\theta(s, a^1, a^2) = \nabla_\theta \log \pi_\theta(a^1 \mid s) - \nabla_\theta \log \pi_\theta(a^2 \mid s) = -\nabla_\theta h_\theta(s, a^2, a^1);$$
$$\nabla^2_\theta h_\theta(s, a^1, a^2) = \nabla^2_\theta \log \pi_\theta(a^1 \mid s) - \nabla^2_\theta \log \pi_\theta(a^2 \mid s) = -\nabla^2_\theta h_\theta(s, a^2, a^1).$$

Now, using the above observations, we can simplify $\nabla^2_\theta l_z(\theta)$ as follows

$$\nabla^2_\theta l_z(\theta) = -y\nabla^2_\theta \log \sigma(\beta h_\theta(s, a^1, a^2)) - (1 - y)\nabla^2_\theta \log \sigma(\beta h_\theta(s, a^2, a^1))$$
$$= -y\big[\beta\sigma(-\beta h_\theta(s, a^1, a^2))\nabla^2_\theta h_\theta(s, a^1, a^2)$$
$$- \beta^2\sigma(\beta h_\theta(s, a^1, a^2))\sigma(-\beta h_\theta(s, a^1, a^2))\nabla_\theta h_\theta(s, a^1, a^2)\nabla_\theta h_\theta(s, a^1, a^2)^\top\big]$$
$$- (1 - y)\big[\beta\sigma(-\beta h_\theta(s, a^2, a^1))\nabla^2_\theta h_\theta(s, a^2, a^1)$$
$$- \beta^2\sigma(\beta h_\theta(s, a^2, a^1))\sigma(-\beta h_\theta(s, a^2, a^1))\nabla_\theta h_\theta(s, a^2, a^1)\nabla_\theta h_\theta(s, a^2, a^1)^\top\big]$$
$$= -y\beta\sigma(-\beta h_\theta(s, a^1, a^2))\nabla^2_\theta h_\theta(s, a^1, a^2)$$
$$+ y\beta^2\sigma(\beta h_\theta(s, a^1, a^2))\sigma(-\beta h_\theta(s, a^1, a^2))\nabla_\theta h_\theta(s, a^1, a^2)\nabla_\theta h_\theta(s, a^1, a^2)^\top$$
$$- (1 - y)\beta\sigma(-\beta h_\theta(s, a^2, a^1))\nabla^2_\theta h_\theta(s, a^2, a^1)$$
$$+ (1 - y)\beta^2\sigma(\beta h_\theta(s, a^2, a^1))\sigma(-\beta h_\theta(s, a^2, a^1))\nabla_\theta h_\theta(s, a^2, a^1)\nabla_\theta h_\theta(s, a^2, a^1)^\top$$
$$\overset{(a)}{=} -y\beta\sigma(-\beta h_\theta(s, a^1, a^2))\nabla^2_\theta h_\theta(s, a^1, a^2)$$
$$+ y\beta^2\sigma(\beta h_\theta(s, a^1, a^2))\sigma(-\beta h_\theta(s, a^1, a^2))\nabla_\theta h_\theta(s, a^1, a^2)\nabla_\theta h_\theta(s, a^1, a^2)^\top$$
$$+ (1 - y)\beta\sigma(-\beta h_\theta(s, a^2, a^1))\nabla^2_\theta h_\theta(s, a^1, a^2)$$
$$+ (1 - y)\beta^2\sigma(-\beta h_\theta(s, a^1, a^2))\sigma(\beta h_\theta(s, a^1, a^2))\nabla_\theta h_\theta(s, a^1, a^2)\nabla_\theta h_\theta(s, a^1, a^2)^\top$$
$$= \beta(-y + \sigma(\beta h_\theta(s, a^1, a^2)))\nabla^2_\theta h_\theta(s, a^1, a^2)$$
$$+ \beta^2\sigma(\beta h_\theta(s, a^1, a^2))\sigma(-\beta h_\theta(s, a^1, a^2))\nabla_\theta h_\theta(s, a^1, a^2)\nabla_\theta h_\theta(s, a^1, a^2)^\top.$$

where $(a)$ is due to $h_\theta(s, a^2, a^1) = -h_\theta(s, a^1, a^2)$, $\nabla_\theta h_\theta(s, a^2, a^1) = -\nabla_\theta h_\theta(s, a^1, a^2)$ and $\nabla^2_\theta h_\theta(s, a^2, a^1) = -\nabla^2_\theta h_\theta(s, a^1, a^2)$. It's clear that we have to calculate $\nabla^2_\theta h_\theta(s, a^1, a^2)$ and $\nabla_\theta h_\theta(s, a^1, a^2)$. Observe that

$$\nabla_\theta h_\theta(s, a^1, a^2) = \nabla_\theta \log \pi_\theta(a^1 \mid s) - \nabla_\theta \log \pi_\theta(a^2 \mid s) = \frac{1}{\pi_\theta(a^1 \mid s)}\nabla_\theta \pi_\theta(a^1 \mid s) - \frac{1}{\pi_\theta(a^2 \mid s)}\nabla_\theta \pi_\theta(a^2 \mid s).$$
$$\tag{15}$$

In addition, we have that $\nabla^2_\theta h_\theta(s, a^1, a^2) = \nabla^2_\theta \log \pi_\theta(a^1 \mid s) - \nabla^2_\theta \log \pi_\theta(a^2 \mid s)$. Using the Hessian chain rule (Eq. (14)), we have

$$\nabla^2_\theta \log \pi_\theta(a \mid s) = \frac{1}{\pi_\theta(a \mid s)}\nabla^2_\theta \pi_\theta(a \mid s) - \frac{1}{\pi_\theta(a \mid s)^2}\nabla_\theta \pi_\theta(a \mid s)\nabla_\theta \pi_\theta(a \mid s)^\top.$$

Now it boils down to tackling $\nabla_\theta \pi_\theta(a \mid s)$ and $\nabla_\theta^2 \pi_\theta(a \mid s)$. Observe that

$$\nabla_\theta \pi_\theta(a \mid s) = \frac{\nabla_\theta \exp\left(\langle \psi(s,a), \theta \rangle\right) \left[\sum_{a'} \exp\left(\langle \psi(s,a'), \theta \rangle\right)\right] - \left[\sum_{a'} \nabla_\theta \exp\left(\langle \psi(s,a'), \theta \rangle\right)\right] \exp\left(\langle \psi(s,a), \theta \rangle\right)}{\left(\sum_{a'} \exp\left(\langle \psi(s,a'), \theta \rangle\right)\right)^2}$$

$$= \frac{\exp\left(\langle \psi(s,a), \theta \rangle\right)}{\sum_{a'} \exp\left(\langle \psi(s,a'), \theta \rangle\right)} \psi(s,a) - \frac{\exp\left(\langle \psi(s,a), \theta \rangle\right)}{\left(\sum_{a'} \exp\left(\langle \psi(s,a'), \theta \rangle\right)\right)^2} \sum_{a'} \exp\left(\langle \psi(s,a'), \theta \rangle\right) \psi(s,a')$$

$$= \frac{\exp\left(\langle \psi(s,a), \theta \rangle\right)}{\sum_{a'} \exp\left(\langle \psi(s,a'), \theta \rangle\right)} \psi(s,a) - \frac{\exp\left(\langle \psi(s,a), \theta \rangle\right)}{\sum_{a'} \exp\left(\langle \psi(s,a'), \theta \rangle\right)} \sum_{a'} \frac{\exp\left(\langle \psi(s,a), \theta \rangle\right)}{\sum_{a''} \exp\left(\langle \psi(s,a''), \theta \rangle\right)} \psi(s,a')$$

$$= \pi_\theta(a \mid s) \psi(s,a) - \pi_\theta(a \mid s) \sum_{a'} \pi_\theta(a' \mid s) \psi(s,a')$$

$$= \pi_\theta(a \mid s) \left[\psi(s,a) - \sum_{a'} \pi_\theta(a' \mid s) \psi(s,a')\right].$$

Then we have

$$\nabla_\theta h_\theta(s, a^1, a^2) = \frac{1}{\pi_\theta(a^1 \mid s)} \pi_\theta(a^1 \mid s) \left[\psi(s, a^1) - \sum_{a'} \pi_\theta(a' \mid s) \psi(s, a')\right]$$

$$- \frac{1}{\pi_\theta(a^2 \mid s)} \pi_\theta(a^2 \mid s) \left[\psi(s, a^2) - \sum_{a'} \pi_\theta(a' \mid s) \psi(s, a')\right]$$

$$= \psi(s, a^1) - \psi(s, a^2). \tag{16}$$

Notice that $\nabla_\theta h_\theta$ above does not depend on the policy parameter $\theta$. This implies that its Hessian is the zero matrix, i.e., $\nabla_\theta^2 h_\theta(s, a^1, a^2) = \mathbf{0}$. Finally, we have that

$$\nabla_\theta^2 l_z(\theta) = \beta^2 \sigma(\beta h_\theta(s, a^1, a^2)) \sigma(-\beta h_\theta(s, a^1, a^2))(\psi(s, a^1) - \psi(s, a^2))(\psi(s, a^1) - \psi(s, a^2))^\top.$$

Moving from the pointwise loss to the empirical loss, we denote

$$\nabla_\theta^2 l_{\mathcal{D}}(\theta) = \frac{1}{n} \sum_{i=1}^n \beta^2 \sigma(\beta h_\theta(s_i, a_i^1, a_i^2)) \sigma(-\beta h_\theta(s_i, a_i^1, a_i^2))(\psi(s_i, a_i^1) - \psi(s_i, a_i^2))(\psi(s_i, a_i^1) - \psi(s_i, a_i^2))^\top.$$

Now let's focus on the function $\sigma(x)\sigma(-x)$. Our aim is to find a lower bound for this function. Observe that

$$|h_\theta(s, a^1, a^2)| = |(\log \pi_\theta(a^1 \mid s) - \log \pi_\theta(a^2 \mid s)) - (\log \pi_{\mathrm{ref}}(a^1 \mid s) - \log \pi_{\mathrm{ref}}(a^2 \mid s))|$$

$$= |\langle \theta, \psi(s, a^1) - \psi(s, a^2) \rangle - \langle \theta_{\mathrm{ref}}, \psi(s, a^1) - \psi(s, a^2) \rangle|$$

$$= |\langle \theta - \theta_{\mathrm{ref}}, \psi(s, a^1) - \psi(s, a^2) \rangle|$$

$$\overset{(a)}{\leq} \|\theta - \theta_{\mathrm{ref}}\|_2 \|\psi(s, a^1) - \psi(s, a^2)\|_2$$

$$\overset{(b)}{\leq} 4B, \tag{17}$$

where $(a)$ is due to Cauchy-Schwarz inequality. $(b)$ is due to the assumptions $\|\theta\|_2 \leq B$ and $\max_{s,a} \|\psi(s,a)\|_2 \leq 1$. Now this suggests that the input to the function $\sigma(\beta h_\theta(s, a^1, a^2)) \sigma(-\beta h_\theta(s, a^1, a^2))$ is bounded in $[-4\beta B, 4\beta B]$. Since $\sigma(x)\sigma(-x)$ is symmetric and strictly decreasing when $x \in [0, \infty)$, we have that

$$\beta^2 \sigma(\beta h_\theta(s, a^1, a^2)) \sigma(-\beta h_\theta(s, a^1, a^2)) \geq \frac{\beta^2 e^{4\beta B}}{(1 + e^{4\beta B})^2}, \quad \forall \theta \in \Theta. \tag{18}$$

We then have that

$$u^\top \nabla_\theta^2 l_{\mathcal{D}}(\theta) u \geq \frac{\gamma}{n} \|Xu\|_2^2, \quad \forall u \in \mathbb{R}^d,$$

where $\gamma = \frac{\beta^2 e^{4\beta B}}{(1 + e^{4\beta B})^2}$ and $X \in \mathbb{R}^{n \times d}$ has the differencing vector $x_i := \psi(s_i, a_i^1) - \psi(s_i, a_i^2) \in \mathbb{R}^d$ as its $i$-th row. Thus, if we introduce the error vector $\Delta := \theta_n^{\mathrm{dpo}} - \theta^*$, then by the linear approximation theorem (Lemma 3), there exists $\alpha \in [0, 1]$ and $\tilde{\theta} = \alpha \theta_n^{\mathrm{dpo}} + (1 - \alpha)\theta^*$ such that

$$l_{\mathcal{D}}(\theta^* + \Delta) - l_{\mathcal{D}}(\theta^*) - \langle \nabla_\theta l_{\mathcal{D}}(\theta^*), \Delta \rangle = \frac{1}{2} \Delta^\top \nabla_\theta^2 l_{\mathcal{D}}(\tilde{\theta}) \Delta \geq \frac{\gamma}{2n} \|X\Delta\|_2^2 = \frac{\gamma}{2} \|\Delta\|_{\Sigma_{\mathcal{D}}}^2, \tag{19}$$

where $\Sigma_{\mathcal{D}} = \frac{1}{n}\sum_{i=1}^{n}(\psi(s_i, a_i^1) - \psi(s_i, a_i^2))(\psi(s_i, a_i^1) - \psi(s_i, a_i^2))^\top$. This implies that $l_{\mathcal{D}}$ is (almost) strongly convex around $\theta^*$ with parameter $\gamma$ with respect to **semi-norm** $\|\cdot\|_{\Sigma_{\mathcal{D}}}$. Note that we will not treat $l_{\mathcal{D}}$ as a strictly strongly convex function in any part of this proof. We only need the inequality Eq. (19).

**Bounding the estimation error.** Recall that $\theta_n^{\mathrm{dpo}}$ is optimal for $l_{\mathcal{D}}(\theta)$ and $\Delta := \theta_n^{\mathrm{dpo}} - \theta^*$. We must have $l_{\mathcal{D}}(\theta_n^{\mathrm{dpo}}) \leq l_{\mathcal{D}}(\theta^*)$. By substracting and adding $\langle \nabla_\theta l_{\mathcal{D}}(\theta^*), \Delta \rangle$ on both sides, we have

$$l_{\mathcal{D}}(\theta^* + \Delta) - l_{\mathcal{D}}(\theta^*) - \langle \nabla_\theta l_{\mathcal{D}}(\theta^*), \Delta \rangle \leq -\langle \nabla_\theta l_{\mathcal{D}}(\theta^*), \Delta \rangle.$$

For the right hand side above, we have

$$|\langle \nabla_\theta l_{\mathcal{D}}(\theta^*), \Delta \rangle| \leq \|\nabla_\theta l_{\mathcal{D}}(\theta^*)\|_{(\Sigma_{\mathcal{D}} + \lambda I)^{-1}} \|\Delta\|_{\Sigma_{\mathcal{D}} + \lambda I}, \quad \text{for any } \lambda > 0.$$

By $\gamma$-strong convexity of $l_{\mathcal{D}}$ at $\theta^*$, we have

$$l_{\mathcal{D}}(\theta^* + \Delta) - l_{\mathcal{D}}(\theta^*) - \langle \nabla_\theta l_{\mathcal{D}}(\theta^*), \Delta \rangle \geq \frac{\gamma}{2}\|\Delta\|_{\Sigma_{\mathcal{D}}}^2.$$

Combining the inequalities, we have $\frac{\gamma}{2}\|\Delta\|_{\Sigma_{\mathcal{D}}}^2 \leq \|\nabla_\theta l_{\mathcal{D}}(\theta^*)\|_{(\Sigma_{\mathcal{D}} + \lambda I)^{-1}} \|\Delta\|_{\Sigma_{\mathcal{D}} + \lambda I}$. Now we need to bound the term $\|\nabla_\theta l_{\mathcal{D}}(\theta^*)\|_{(\Sigma_{\mathcal{D}} + \lambda I)^{-1}}$. We can calculate the gradient w.r.t. $\theta$ of the pointwise loss as follows

$$\begin{aligned}
\nabla_\theta l_z(\theta) &= \nabla_\theta[-y\log\sigma(\beta h_\theta(s, a^1, a^2)) - (1-y)\log\sigma(\beta h_\theta(s, a^2, a^1))] \\
&= -y\nabla_\theta\log\sigma(\beta h_\theta(s, a^1, a^2)) - (1-y)\nabla_\theta\log\sigma(\beta h_\theta(s, a^2, a^1)) \\
&= -\beta y\sigma(-\beta h_\theta(s, a^1, a^2))\nabla_\theta h_\theta(s, a^1, a^2) - \beta(1-y)\sigma(\beta h_\theta(s, a^1, a^2))\nabla_\theta h_\theta(s, a^2, a^1) \\
&\overset{(a)}{=} -\beta(y\sigma(\beta h_\theta(s, a^2, a^1)) - (1-y)\sigma(\beta h_\theta(s, a^1, a^2)))(\psi(s, a^1) - \psi(s, a^2)),
\end{aligned}$$

where $(a)$ is due to $\nabla_\theta h_\theta(s, a^1, a^2) = \psi(s, a^1) - \psi(s, a^2)$ calculated in Eq. (16). This implies that

$$\nabla_\theta l_{\mathcal{D}}(\theta^*) = \frac{-\beta}{n}\sum_{i=1}^{n}[y_i\sigma(\beta h_{\theta^*}(s_i, a_i^2, a_i^1)) - (1-y_i)\sigma(\beta h_{\theta^*}(s_i, a_i^1, a_i^2))]x_i, \tag{20}$$

where $x_i = \psi(s_i, a_i^1) - \psi(s_i, a_i^2)$. Now let's define a random vector $V \in \mathbb{R}^n$ with i.i.d. components as

$$V_i = \begin{cases} \sigma(\beta h_{\theta^*}(s_i, a_i^2, a_i^1)) & \text{w.p. } \sigma(\beta h_{\theta^*}(s_i, a_i^1, a_i^2)), \\ -\sigma(\beta h_{\theta^*}(s_i, a_i^1, a_i^2)) & \text{w.p. } \sigma(\beta h_{\theta^*}(s_i, a_i^2, a_i^1)). \end{cases} \tag{21}$$

Then we have $\nabla_\theta l_{\mathcal{D}}(\theta^*) = -\frac{\beta}{n}X^\top V$. It's easy to verify that $\mathbb{E}V_i = 0$ and $|V_i| \leq 1$, for all $1 \leq i \leq n$. Next, if we define the $n \times n$ matrix $M := \frac{\beta^2}{n^2}X(\Sigma_{\mathcal{D}} + \lambda I)^{-1}X^\top$, then we can write $\|\nabla_\theta l_{\mathcal{D}}(\theta^*)\|_{(\Sigma_{\mathcal{D}} + \lambda I)^{-1}}^2 = V^\top MV$. Let the eigendecomposition of $X^\top X$ be $U\Lambda U^\top$. Observe that

$$M = \frac{\beta^2}{n^2}X(\Sigma_{\mathcal{D}} + \lambda I)^{-1}X^\top = \frac{\beta^2}{n^2}XU(\Lambda/n + \lambda I)^{-1}U^\top X^\top.$$

We can bound the trace of $M$ as follows

$$\begin{aligned}
\mathsf{Tr}(M) &= \mathsf{Tr}(\frac{\beta^2}{n^2}XU(\Lambda/n + \lambda I)^{-1}U^\top X^\top) = \frac{\beta^2}{n^2}\mathsf{Tr}(U(\Lambda/n + \lambda I)^{-1}U^\top U\Lambda U^\top) \\
&= \frac{\beta^2}{n^2}\mathsf{Tr}(U(\Lambda/n + \lambda I)^{-1}\Lambda U^\top) = \frac{\beta^2}{n^2}\mathsf{Tr}((\Lambda/n + \lambda I)^{-1}\Lambda) = \frac{\beta^2}{n^2}\sum_{i=1}^{d}\frac{ne_i}{e_i + \lambda n} \\
&\leq \frac{\beta^2}{n^2}\cdot nd = \frac{\beta^2 d}{n},
\end{aligned}$$

where $e_i$ is the $i$-th eigenvalue of $X^\top X$. Similarly, we can bound $\mathsf{Tr}(M^2) \leq \frac{\beta^4 d}{n^2}$. Now, let $X = \widetilde{U}\Sigma\widetilde{V}^\top$ be the singular value decomposition of $X$. Then we can show that

$$M = \frac{\beta^2}{n^2}X(X^\top X/n + \lambda I)^{-1}X^\top = \frac{\beta^2}{n^2}\widetilde{U}\Sigma(\Sigma^\top\Sigma/n + \lambda I)^{-1}\Sigma\widetilde{U}^\top.$$

Since $X(\Sigma_{\mathcal{D}} + \lambda I)^{-1}X^\top$ is symmetric, and clearly $\widetilde{U}\Sigma(\Sigma^\top\Sigma/n + \lambda I)^{-1}\Sigma\widetilde{U}^\top$ diagonalizes it, the eigenvalue of it takes form $\frac{\sigma_i^2}{\sigma_i^2/n + \lambda}$, where $\sigma_i$ is the $i$-th singular value of $X$. Hence, all eigenvalues

are upper bounded by $n$. Then we must have $\|M\|_{\mathrm{op}} = \lambda_{\max}(M) \le \frac{\beta^2}{n}$. Since the components of $V$ are i.i.d. with $\mathbb{E}V_i = 0$ and $|V_i| \le 1$, the elements are 1-sub-Gaussian, we can use the Bernstein's inequality for sub-Gaussian random variables in quadratic form (see Lemma 8). It implies that with probability at least $1 - \delta$,

$$\|\nabla_\theta l_{\mathcal{D}}(\theta^*)\|_{(\Sigma_{\mathcal{D}}+\lambda I)^{-1}}^2 = V^\top M V \le \mathsf{Tr}(M) + 2\sqrt{\mathsf{Tr}(M^2)\log(1/\delta)} + 2\|M\|_{\mathrm{op}}\log(1/\delta)$$

$$\le \frac{\beta^2 d}{n} + 2\sqrt{\frac{\beta^4}{n^2}d\log(1/\delta)} + 2\frac{\beta^2}{n}\log(1/\delta) = \frac{\beta^2}{n}(d + 2\sqrt{d\log(1/\delta)} + 2\log(1/\delta)).$$

Set $a = \sqrt{d}$ and $b = \sqrt{\log(1/\delta)}$. Note that we have

$$d + 2\sqrt{d\log(1/\delta)} + 2\log(1/\delta) = (a+b)^2 + b^2$$
$$\le 2(a+b)^2 = 2(a^2 + b^2 + 2ab)$$
$$\le 2(a^2 + b^2 + a^2 + b^2) = 4(a^2 + b^2) = 4(d + \log(1/\delta)),$$

where the last inequality is due to AM-GM inequality. Altogether, we have $\|\nabla_\theta l_{\mathcal{D}}(\theta^*)\|_{(\Sigma_{\mathcal{D}}+\lambda I)^{-1}}^2 \le \frac{4\beta^2}{n}(d + \log(1/\delta))$.

The final assembly now begins as follows

$$\frac{\gamma}{2}\|\Delta\|_{\Sigma_{\mathcal{D}}+\lambda I}^2 = \frac{\gamma}{2}\|\Delta\|_{\Sigma_{\mathcal{D}}}^2 + \frac{\gamma}{2}\|\Delta\|_{\lambda I}^2 = \frac{\gamma}{2}\|\Delta\|_{\Sigma_{\mathcal{D}}}^2 + \frac{\lambda\gamma}{2}\|\Delta\|^2$$

$$\le \|\nabla_\theta l_{\mathcal{D}}(\theta^*)\|_{(\Sigma_{\mathcal{D}}+\lambda I)^{-1}}\|\Delta\|_{\Sigma_{\mathcal{D}}+\lambda I} + \frac{\lambda\gamma}{2}\|\Delta\|^2$$

$$\le \sqrt{\frac{4\beta^2}{n}(d + \log(1/\delta))}\|\Delta\|_{\Sigma_{\mathcal{D}}+\lambda I} + \frac{\lambda\gamma}{2}4B^2,$$

where the last inequality uses triangle inequality and the assumption that $\|\theta\| \le B, \forall \theta \in \Theta$. This implies that

$$\|\Delta\|_{\Sigma_{\mathcal{D}}+\lambda I}^2 \le \frac{2}{\gamma}\sqrt{\frac{4\beta^2}{n}(d + \log(1/\delta))}\|\Delta\|_{\Sigma_{\mathcal{D}}+\lambda I} + 4\lambda B^2.$$

Now denote $\alpha = \frac{2}{\gamma}\sqrt{\frac{4\beta^2}{n}(d + \log(1/\delta))}$ and $\beta = 4\lambda B^2$, and let $x = \|\Delta\|_{\Sigma_{\mathcal{D}}+\lambda I}$. Since we have $x^2 - \alpha x - \beta \le 0$, then $x$ must be less than the bigger root, i.e.,

$$x \le \frac{\alpha + \sqrt{\alpha^2 + 4\beta}}{2} \le \sqrt{\frac{\alpha^2 + \alpha^2 + 4\beta}{2}} = \sqrt{\alpha^2 + 2\beta},$$

where the second inequality is by Jensen's inequality. Finally, we have that

$$\|\theta_n^{\mathrm{dpo}} - \theta^*\|_{\Sigma_{\mathcal{D}}+\lambda I} = \|\Delta\|_{\Sigma_{\mathcal{D}}+\lambda I} \le 2\sqrt{\frac{4\beta^2}{\gamma^2 n}(d + \log(1/\delta)) + 2\lambda B^2}.$$

### B.2  Proof of WDPO Loss Function Convergence

**Lemma 9** (Convergence of WDPO loss). *Fix any $\theta \in \Theta$ and $\rho > 0$. Let $\delta \in (0, 1)$. With probability $1 - \delta$,*

$$|\mathcal{L}^{\mathrm{W}}(\theta; \rho) - \mathcal{L}_n^{\mathrm{W}}(\theta; \rho)| \le \sqrt{\frac{K^2 \log(2/\delta)}{2n}},$$

*where $K = |\log\sigma(-4\beta B)|$.*

*Proof.* Recall the strong duality in Lemma 1. The term $\inf_{z \in \mathcal{Z}}[\eta d^p(z, z') - l(z; \theta)]$ is called the *Moreau-Yosida regularization* of $-l$ with parameter $1/\eta$. We denote it by $l_\eta(z; \theta)$. Now observe that

$$\left|\mathcal{L}^{\mathrm{W}}(\theta; \rho) - \mathcal{L}_n^{\mathrm{W}}(\theta; \rho)\right| = \left|\sup_{\mathsf{P}:\, W_p(\mathsf{P},\mathsf{P}^o)\le\rho} \mathbb{E}_{z\sim\mathsf{P}}[l_z(\theta)] - \sup_{\mathsf{P}:\, W_p(\mathsf{P},\mathsf{P}_n^o)\le\rho} \mathbb{E}_{z\sim\mathsf{P}}[l_z(\theta)]\right|$$

$$\overset{(a)}{=} \left|\inf_{\eta\ge 0}\{\eta\rho^p - \mathbb{E}_{z\sim\mathsf{P}^o}[l_\eta(z; \theta)]\} - \inf_{\eta\ge 0}\{\eta\rho^p - \mathbb{E}_{z\sim\mathsf{P}_n^o}[l_\eta(z; \theta)]\}\right|$$

$$\overset{(b)}{\le} \sup_{\eta\ge 0}\left|\mathbb{E}_{z\sim\mathsf{P}^o}[l_\eta(z; \theta)] - \mathbb{E}_{z\sim\mathsf{P}_n^o}[l_\eta(z; \theta)]\right|,$$

where $(a)$ is by the strong duality, and $(b)$ is due to $|\inf_x f(x) - \inf_x g(x)| \le \sup_x |f(x) - g(x)|$. Next, we will show that, for any $\eta \ge 0$, the function $l_\eta$ is a bounded function. We first prove its upper bound. The negative DPO loss takes the following form:

$$-l(z;\theta) = y\log\sigma(x) + (1-y)\log\sigma(-x) \le 0, \quad y \in \{0,1\}.$$

The inequality is because the sigmoid function is *strictly* bounded between 0 and 1, i.e., $\sigma \in (0,1)$. This implies that $\log\sigma$ is non-positive. Using this, we have that

$$l_\eta(z;\theta) = \inf_{z' \in \mathcal{Z}}[\eta d^p(z',z) - l(z';\theta)] \le \inf_{z' \in \mathcal{Z}}[\eta d^p(z',z)] = 0.$$

Now we prove its lower bound. Recall that in the analysis of non-robust DPO loss, we proved that $|h_\theta(s,a^1,a^2)| \le 4B$ (see Eq. (17)). Since both $\log$ and $\sigma$ are increasing functions, we have that $\log\sigma(\beta h_\theta(s,a^1,a^2)) \ge \log\sigma(-4\beta B)$. Now observe that

$$\begin{aligned}
l_\eta(z;\theta) &= \inf_{z' \in \mathcal{Z}}[\eta d^p(z',z) - l(z;'\theta)]\\
&\ge \inf_{z' \in \mathcal{Z}}[-l(z';\theta)] = \inf_{s,a^1,a^2,y}[y\log\sigma(\beta h_\theta(s,a^1,a^2)) + (1-y)\log\sigma(\beta h_\theta(s,a^2,a^1))]\\
&\ge \log\sigma(-4\beta B),
\end{aligned}$$

where the first inequality is because both $\eta$ and metric $d^p$ are non-negative. The last inequality is because only one of the $\log\sigma$ term will be activated and the lower bound we recalled above. Denote $K = |\log\sigma(-4\beta B)|$. Since $l_\eta$ is a bounded function, by Hoeffding's inequality for bounded random variable (Lemma 7), we have

$$\mathbb{P}\left(\left|\mathbb{E}_{z\sim\mathsf{P}^o}[l_\eta(z;\theta)] - \mathbb{E}_{z\sim\mathsf{P}_n^o}[l_\eta(z;\theta)]\right| \ge \epsilon\right) \le 2\exp\left(\frac{-2n\epsilon^2}{K^2}\right).$$

By picking $\delta$ to be the right hand side above, we have that, with probability at least $1 - \delta$,

$$|\mathbb{E}_{z\sim\mathsf{P}^o}[l_\eta(z;\theta)] - \mathbb{E}_{z\sim\mathsf{P}_n^o}[l_\eta(z;\theta)]| \le \sqrt{\frac{K^2\log(2/\delta)}{2n}}.$$

Since $K$ does not depend on $\eta$, such concentration is uniform for all functions $l_\eta, \eta \ge 0$. We have the desired result. $\qquad\square$

### B.3 Proof of the Strong Convexity of WDPO Loss

We first prove that the function $g(\theta;\mathsf{P}) := \mathbb{E}_{z\sim\mathsf{P}}[l(z;\theta)]$ is strongly convex, for any $\mathsf{P}$, as follows:

**Lemma 10.** *Let $l(z;\theta)$ be the DPO loss function. Assume that Assumption 2 is in place. Then $g(\theta) := \mathbb{E}_{z\sim\mathsf{P}}[l(z;\theta)]$ is $\gamma$-strongly convex with respect to norm $\|\cdot\|_{\Sigma_\mathsf{P}}$, where $\Sigma_\mathsf{P} = \mathbb{E}_{(s,a^1,a^2,y)\sim\mathsf{P}}(\psi(s,a^1) - \psi(s,a^2))(\psi(s,a^1) - \psi(s,a^2))^\top$, and $\gamma = \frac{\beta^2 e^{4\beta B}}{(1+e^{4\beta B})^2}$.*

*Proof.* Recall that we proved that the Hessian of the pointwise DPO loss takes the form:

$$\nabla_\theta^2 l_z(\theta) = \beta^2\sigma(\beta h_\theta(s,a^1,a^2))\sigma(-\beta h_\theta(s,a^1,a^2))(\psi(s,a^1) - \psi(s,a^2))(\psi(s,a^1) - \psi(s,a^2))^\top.$$

In addition, we also proved that (see Eq. (18))

$$\beta^2\sigma(\beta h_\theta(s,a^1,a^2))\sigma(-\beta h_\theta(s,a^1,a^2)) \ge \frac{\beta^2 e^{4\beta B}}{(1+e^{4\beta B})^2}, \quad \forall\theta \in \Theta.$$

This implies that

$$u^\top\nabla_\theta^2 l_z(\theta)u \ge \gamma\|(\psi(s,a^1) - \psi(s,a^2))^\top u\|_2^2, \quad \forall u \in \mathbb{R}^d,$$

where $\gamma = \frac{\beta^2 e^{4\beta B}}{(1+e^{4\beta B})^2}$. Thus, if we introduce the error vector $\Delta := \theta' - \theta$, where $\theta, \theta' \in \Theta$, then by the linear approximation theorem (Lemma 3), there exists $\alpha \in [0,1]$ and $\tilde\theta = \alpha\theta + (1-\alpha)\theta'$ such that

$$l_z(\theta+\Delta) - l_z(\theta) - \langle\nabla_\theta l_z(\theta),\Delta\rangle = \frac{1}{2}\Delta^\top\nabla_\theta^2 l_z(\tilde\theta)\Delta \ge \frac{\gamma}{2}\|(\psi(s,a^1)-\psi(s,a^2))^\top\Delta\|_2^2 = \frac{\gamma}{2}\|\Delta\|_{\Sigma_z}^2,$$

$$\tag{22}$$

where $\Sigma_z = (\psi(s, a^1) - \psi(s, a^2))(\psi(s, a^1) - \psi(s, a^2))^\top$. Note that $\Sigma_z$ is only semi-definite. Let $\alpha \in [0, 1]$ and $\theta, \theta' \in \Theta$. Observe that

$$
\begin{aligned}
g(\alpha\theta + (1-\alpha)\theta') &= \mathbb{E}_{z\sim\mathsf{P}}[l(\alpha\theta + (1-\alpha)\theta'; z)] \\
&\overset{(a)}{\leq} \mathbb{E}_{z\sim\mathsf{P}}\left[\alpha l(z; \theta) + (1-\alpha)l(\theta'; z) - \frac{\gamma}{2}\alpha(1-\alpha)\|\theta - \theta'\|_{\Sigma_z}^2\right] \\
&= \alpha g(\theta) + (1-\alpha)g(\theta') - \frac{\gamma}{2}\alpha(1-\alpha)(\theta - \theta')^\top \mathbb{E}_{\mathsf{P}}[\Sigma_z](\theta - \theta') \\
&= \alpha g(\theta) + (1-\alpha)g(\theta') - \frac{\gamma}{2}\alpha(1-\alpha)\|\theta - \theta'\|_{\Sigma_{\mathsf{P}}}^2,
\end{aligned}
$$

where $(a)$ is by Lemma 4. In particular, the equivalence between the inequalities, Eq. (22) and $(a)$, can be found in the proof of Beck (2017, Theorem 5.24), and the author would like to comment that the proof does not rely on whether $\|\cdot\|_{\Sigma_z}$ is a semi-norm or a norm. Now, by Assumption 2, $\Sigma_{\mathsf{P}}$ is strictly positive definite, hence $\|\cdot\|_{\Sigma_{\mathsf{P}}}$ is a norm. This implies that $g$ is $\gamma$-strongly convex with respect to $\|\cdot\|_{\Sigma_{\mathsf{P}}}$. $\qquad\square$

Now, we are ready to prove our main strong convexity lemma.

**Lemma 11.** *Let $l(z; \theta)$ be the DPO loss function. The Wasserstein distributionally robust DPO loss function,*

$$
\mathcal{L}^{\mathrm{W}}(\theta; \rho) := \sup_{\mathsf{P}:\, W_p(\mathsf{P}, \mathsf{P}^o) \leq \rho} \mathbb{E}_{z\sim\mathsf{P}}[l(z; \theta)],
$$

*is $\gamma\lambda$-strongly convex in $\theta$ with respect to (non-weighted) 2-norm $\|\cdot\|_2$, where $\lambda$ is the regularity condition number defined in Assumption 2, and $\gamma = \frac{\beta^2 e^{4\beta B}}{(1+e^{4\beta B})^2}$.*

*Proof.* Let $\alpha \in [0, 1]$ and $\theta, \theta' \in \Theta$. First, we denote $h(\theta; \mathsf{P}) = \mathbb{E}_{z\sim\mathsf{P}}[l(z; \theta)]$ for any $\mathsf{P}$ in the Wasserstein ball. In Lemma 10, we proved that $h$ is $\gamma$-strongly convex in $\theta$ w.r.t. norm $\|\cdot\|_{\Sigma_{\mathsf{P}}}$. Now observe that

$$
\begin{aligned}
\mathcal{L}^{\mathrm{W}}(\alpha\theta + (1-\alpha)\theta'; \rho) &= \sup_{\mathsf{P}:\, W_p(\mathsf{P}, \mathsf{P}^o) \leq \rho} h(\alpha\theta + (1-\alpha)\theta'; z) \\
&\overset{(a)}{\leq} \sup_{\mathsf{P}:\, W_p(\mathsf{P}, \mathsf{P}^o) \leq \rho} \left\{\alpha h(\theta; \mathsf{P}) + (1-\alpha)h(\theta'; \mathsf{P}) - \frac{\gamma}{2}\alpha(1-\alpha)\|\theta - \theta'\|_{\Sigma_{\mathsf{P}}}^2\right\} \\
&\overset{(b)}{\leq} \alpha\mathcal{L}^{\mathrm{W}}(\theta; \rho) + (1-\alpha)\mathcal{L}^{\mathrm{W}}(\theta'; \rho) + \sup_{\mathsf{P}:\, W_p(\mathsf{P}, \mathsf{P}^o) \leq \rho} -\frac{\gamma}{2}\alpha(1-\alpha)\|\theta - \theta'\|_{\Sigma_{\mathsf{P}}}^2 \\
&= \alpha\mathcal{L}^{\mathrm{W}}(\theta; \rho) + (1-\alpha)\mathcal{L}^{\mathrm{W}}(\theta'; \rho) - \frac{\gamma}{2}\alpha(1-\alpha)\inf_{\mathsf{P}:\, W_p(\mathsf{P}, \mathsf{P}^o) \leq \rho}\|\theta - \theta'\|_{\Sigma_{\mathsf{P}}}^2 \\
&\leq \alpha\mathcal{L}^{\mathrm{W}}(\theta; \rho) + (1-\alpha)\mathcal{L}^{\mathrm{W}}(\theta'; \rho) - \frac{\gamma}{2}\alpha(1-\alpha)\inf_{\mathsf{P}:\, W_p(\mathsf{P}, \mathsf{P}^o) \leq \rho}\lambda_{\min}(\Sigma_{\mathsf{P}})\|\theta - \theta'\|_2^2 \\
&\overset{(c)}{\leq} \alpha\mathcal{L}^{\mathrm{W}}(\theta; \rho) + (1-\alpha)\mathcal{L}^{\mathrm{W}}(\theta'; \rho) - \frac{\gamma\lambda}{2}\alpha(1-\alpha)\|\theta - \theta'\|_2^2.
\end{aligned}
$$

Note that the function $g(\theta) = \mathbb{E}_{z\sim\mathsf{P}}[l(z; \theta)]$ is $\gamma$-strongly convex with respect to $\|\cdot\|_{\Sigma_{\mathsf{P}}}$ by Lemma 10. We use this fact in $(a)$. The inequality in $(b)$ is due to $\sup_x(f(x) + g(x)) \leq \sup_x f(x) + \sup_x g(x)$. The last inequality $(c)$ is because $\lambda_{\min}(\Sigma_{\mathsf{P}}) \geq \lambda$, for all $\mathsf{P} \in \mathcal{P}_{\mathsf{W}}$ by Assumption 2. This implies that $\mathcal{L}^{\mathrm{W}}$ is a $\gamma\lambda$-strongly convex function with respect to $\|\cdot\|_2$. $\qquad\square$

### B.4 Proof of Policy Parameter Convergence of WDPO

By Lemma 9, we have that, with probability at least $1 - \delta$,

$$
\begin{aligned}
\mathcal{L}^{\mathrm{W}}(\theta_n^{\mathrm{W}}; \rho) &- \mathcal{L}^{\mathrm{W}}(\theta^{\mathrm{W}}; \rho) \\
&= \mathcal{L}^{\mathrm{W}}(\theta_n^{\mathrm{W}}; \rho) - \mathcal{L}_n^{\mathrm{W}}(\theta_n^{\mathrm{W}}; \rho) + \mathcal{L}_n^{\mathrm{W}}(\theta_n^{\mathrm{W}}; \rho) - \mathcal{L}_n^{\mathrm{W}}(\theta^{\mathrm{W}}; \rho) + \mathcal{L}_n^{\mathrm{W}}(\theta^{\mathrm{W}}; \rho) - \mathcal{L}^{\mathrm{W}}(\theta^{\mathrm{W}}; \rho) \\
&\leq |\mathcal{L}^{\mathrm{W}}(\theta_n^{\mathrm{W}}; \rho) - \mathcal{L}_n^{\mathrm{W}}(\theta_n^{\mathrm{W}}; \rho)| + |\mathcal{L}_n^{\mathrm{W}}(\theta^{\mathrm{W}}; \rho) - \mathcal{L}^{\mathrm{W}}(\theta^{\mathrm{W}}; \rho)| \\
&\leq \sqrt{\frac{2K^2 \log(2/\delta)}{n}},
\end{aligned}
$$

where the first inequality is because $\theta_n^{\mathrm{W}}$ is the minimizer of $\mathcal{L}_n^{\mathrm{W}}$. Now by the $\gamma\lambda$-strong convexity of $\mathcal{L}^{\mathrm{W}}$ (see Lemma 11) and Lemma 5.II, we have that

$$\|\theta_n^{\mathrm{W}} - \theta^{\mathrm{W}}\|_2^2 \le \sqrt{\frac{8K^2 \log(2/\delta)}{\gamma^2 \lambda^2 n}}.$$

## C   Proof of KLDPO Sample Complexity

We state a result from Hu and Hong (2013) that proves an equivalent condition for the infimum to be achieved at $\lambda^* = 0$.

**Proposition 3** (Hu and Hong, 2013, Proposition 2). *Let $l_u(z; \theta)$ be the essential supremum of $l(z; \theta)$ under measure $\mathsf{P}^o$, i.e.,*

$$l_u(z; \theta) = \inf\{t \in \mathbb{R} \colon \mathbb{P}(l(z; \theta) > t) = 0\}.$$

*Also let $\kappa_u = \mathbb{P}(l(z; \theta) = l_u(z; \theta))$, i.e., $\kappa_u$ is the probability mass of the distribution $\mathsf{P}^o$ on the essential supremum of $l$. Then $\lambda^* = 0$ if and only if $l_u(z; \theta) < +\infty$, $\kappa_u > 0$, and $\log \kappa_u + \rho \ge 0$, where $\rho$ is the diameter of the KL uncertainty set.*

We now make an assumption on the loss function(s) $l(\cdot; \theta)$, $\theta \in \Theta$. Note that this assumption is only used in proving the dual reformulation of KLDPO objective.

**Assumption 3.** *We assume that $l(z; \theta) \le L$ for all $\theta \in \Theta$. That is, the loss function is upper bounded by $L$. In addition, we also assume that $\Theta$ permits a uniform upper bound on $\lambda_\theta$. That is, we assume that $\sup_{\theta \in \Theta} \lambda_\theta < \overline{\lambda}$.*

We now prove the following dual reformulation result:

**Lemma 12.** *Let $l(z; \theta)$ be the DPO loss. The KLDPO loss function has the following dual reformulation*

$$\mathcal{L}^{\mathrm{KL}}(\theta; \rho) = \sup_{\mathsf{P} \colon D_{\mathrm{KL}}(\mathsf{P} \| \mathsf{P}^o) \le \rho} \mathbb{E}_{z \sim \mathsf{P}}[l(z; \theta)] = \inf_{\lambda \in [\underline{\lambda}, \overline{\lambda}]} \left\{ \lambda\rho + \lambda \log\left( \mathbb{E}_{z \sim \mathsf{P}^o}\left[ \exp\left( \frac{l(z; \theta)}{\lambda} \right) \right] \right) \right\},$$

*where $0 < \underline{\lambda} < \overline{\lambda} < \infty$ are some constants.*

*Proof.* We include the derivation here for completeness. Previous works in optimization and distributionally robust reinforcement learning have covered the dual problem of distributionally robust optimization with KL uncertainty set (e.g., see Hu and Hong (2013); Panaganti and Kalathil (2022); Xu et al. (2023)).

Recall that $f(t) = t \log(t)$ corresponds to the KL divergence. The optimal $t$ for $f^*(s) = \sup_{t \ge 0}\{st - t\log(t)\}$ is $\exp(s - 1)$. This implies that the Fenchel conjugate of $f$ is $f^*(s) = \exp(s - 1)$. From Lemma 6, we get

$$\sup_{\mathsf{P} \colon D_{\mathrm{KL}}(\mathsf{P} \| \mathsf{P}^o) \le \rho} \mathbb{E}_{z \sim \mathsf{P}}[l(z; \theta)] = \inf_{\lambda \ge 0, \eta \in \mathbb{R}} \left\{ \mathbb{E}_{z \sim \mathsf{P}^o}\left[ \lambda f^*\left( \frac{l(z; \theta) - \eta}{\lambda} \right) \right] + \lambda\rho + \eta \right\}$$

$$= \inf_{\lambda \ge 0, \eta \in \mathbb{R}} \left\{ \mathbb{E}_{z \sim \mathsf{P}^o}\left[ \lambda \exp\left( \frac{l(z; \theta) - \eta}{\lambda} - 1 \right) \right] + \lambda\rho + \eta \right\}$$

$$= \inf_{\lambda \ge 0} \left\{ \lambda\rho + \lambda \log\left( \mathbb{E}_{z \sim \mathsf{P}^o}\left[ \exp\left( \frac{l(z; \theta)}{\lambda} \right) \right] \right) \right\},$$

where the last equality by plugging in the optimal $\eta$, i.e., $\eta^* = \lambda \log(\mathbb{E}_{z \sim \mathsf{P}^o}[\exp(l(z; \theta)/\lambda - 1)])$. Now observe that

$$h(\lambda; \theta) := \lambda\rho + \lambda \log\left( \mathbb{E}_{z \sim \mathsf{P}^o}\left[ \exp\left( \frac{l(z; \theta)}{\lambda} \right) \right] \right) \ge \lambda\rho =: g(\lambda).$$

The inequality is because the loss function is non-negative, i.e., $l \ge 0$, and $h$ is increasing in $l$. Now $g(\lambda)$ is a strictly increasing function that lower bounds function $h(\lambda; \theta)$. Since $g(\lambda) \to \infty$ as $\lambda \to \infty$, $h(\lambda; \theta)$ cannot achieve its infimum at $\infty$. In other words, there exists $\overline{\lambda}_\theta$ such that

$$h(\lambda; \theta) \ge g(\lambda) > g(\overline{\lambda}_\theta), \forall \quad \lambda > \overline{\lambda}_\theta.$$

This implies that it suffices to seek the infimum in $[0, \overline{\lambda}_\theta]$. Hence, we have

$$\mathcal{L}^{\mathrm{KL}}(\theta; \rho) = \inf_{\lambda \in [0, \overline{\lambda}_\theta]} \left\{ \lambda \rho + \lambda \log \left( \mathbb{E}_{z \sim \mathsf{P}^o} \left[ \exp \left( \frac{l(z; \theta)}{\lambda} \right) \right] \right) \right\}.$$

Now from Proposition 3, the condition $\log \kappa_u + \rho \geq 0$ is problem-dependent due to the diameter $\rho$, which is a design choice. Note that when $\kappa_u$ is close to zero, the condition $\log \kappa_u + \rho \geq 0$ is almost never true for a reasonable $\rho$. Hence, we ignore the case where $\lambda^* = 0$. By Assumption 3, without loss of generality, we have that $\lambda^* \in [\underline{\lambda}, \overline{\lambda}]$, where $\underline{\lambda}$ is some problem-specific constant. Then we have the result. In the literature of distributionally robust reinforcement learning, similar arguments can be found in Zhou et al. (2021); Panaganti and Kalathil (2022). □

**Lemma 13.** *Fix any $\theta \in \Theta$ and $\rho > 0$. Let $\delta \in (0, 1)$. Assume Assumption 3 is in place. With probability $1 - \delta$, we have that*

$$|\mathcal{L}^{\mathrm{KL}}(\theta; \rho) - \mathcal{L}_n^{\mathrm{KL}}(\theta; \rho)| \leq \overline{\lambda} \sqrt{\frac{\exp\left(L/\underline{\lambda}\right) \log(2/\delta)}{2n}}, \quad \forall \epsilon > 0,$$

*where $\underline{\lambda}, \overline{\lambda}$ are some constants that are independent of $\epsilon$.*

*Proof.* Observe that

$$
|\mathcal{L}^{\mathrm{KL}}(\theta; \rho) - \mathcal{L}_n^{\mathrm{KL}}(\theta; \rho)| = \left| \sup_{\mathsf{P}:\, D_{\mathrm{KL}}(\mathsf{P} \,\|\, \mathsf{P}^o) \leq \rho} \mathbb{E}_{z \sim \mathsf{P}}[l(z; \theta)] - \sup_{\mathsf{P}:\, D_{\mathrm{KL}}(\mathsf{P} \,\|\, \mathsf{P}_n^o) \leq \rho} \mathbb{E}_{z \sim \mathsf{P}}[l(z; \theta)] \right|
$$

$$
\overset{(a)}{=} \left| \inf_{\lambda \in [\underline{\lambda}, \overline{\lambda}]} \left\{ \lambda \rho + \lambda \log \left( \mathbb{E}_{z \sim \mathsf{P}^o} \left[ \exp \left( \frac{l(z; \theta)}{\lambda} \right) \right] \right) \right\} \right.
$$
$$
\left. - \inf_{\lambda \in [\underline{\lambda}, \overline{\lambda}]} \left\{ \lambda \rho + \lambda \log \left( \mathbb{E}_{z \sim \mathsf{P}_n^o} \left[ \exp \left( \frac{l(z; \theta)}{\lambda} \right) \right] \right) \right\} \right|
$$

$$
\overset{(b)}{\leq} \sup_{\lambda \in [\underline{\lambda}, \overline{\lambda}]} \left| \lambda \log \left( \mathbb{E}_{z \sim \mathsf{P}_n^o} \left[ \exp \left( \frac{l(z; \theta)}{\lambda} \right) \right] \right) - \lambda \log \left( \mathbb{E}_{z \sim \mathsf{P}^o} \left[ \exp \left( \frac{l(z; \theta)}{\lambda} \right) \right] \right) \right|
$$

$$
\overset{(c)}{=} \sup_{\lambda \in [\underline{\lambda}, \overline{\lambda}]} \lambda \left| \log \left( \frac{\mathbb{E}_{z \sim \mathsf{P}_n^o} \left[ \exp\left(l(z; \theta)\right)/\lambda \right]}{\mathbb{E}_{z \sim \mathsf{P}^o} \left[ \exp\left(l(z; \theta)\right)/\lambda \right]} \right) \right|
$$

$$
\leq \sup_{\lambda \in [\underline{\lambda}, \overline{\lambda}]} \lambda \left| \log \left( \frac{\left| \mathbb{E}_{z \sim \mathsf{P}_n^o} \left[ \exp\left(l(z; \theta)\right)/\lambda \right] - \mathbb{E}_{z \sim \mathsf{P}^o} \left[ \exp\left(l(z; \theta)\right)/\lambda \right] \right|}{\mathbb{E}_{z \sim \mathsf{P}^o} \left[ \exp\left(l(z; \theta)\right)/\lambda \right]} + 1 \right) \right|
$$

$$
\overset{(d)}{\leq} \sup_{\lambda \in [\underline{\lambda}, \overline{\lambda}]} \lambda \frac{\left| \mathbb{E}_{z \sim \mathsf{P}_n^o} \left[ \exp\left(l(z; \theta)\right)/\lambda \right] - \mathbb{E}_{z \sim \mathsf{P}^o} \left[ \exp\left(l(z; \theta)\right)/\lambda \right] \right|}{\mathbb{E}_{z \sim \mathsf{P}^o} \left[ \exp\left(l(z; \theta)\right)/\lambda \right]}
$$

$$
\overset{(e)}{\leq} \overline{\lambda} \sup_{\lambda \in [\underline{\lambda}, \overline{\lambda}]} \left| \mathbb{E}_{z \sim \mathsf{P}_n^o} \left[ \exp\left(l(z; \theta)\right)/\lambda \right] - \mathbb{E}_{z \sim \mathsf{P}^o} \left[ \exp\left(l(z; \theta)\right)/\lambda \right] \right|,
$$

where $(a)$ is by Lemma 12. $(b)$ is because $|\inf_x f(x) - \inf_x g(x)| \leq \sup_x |f(x) - g(x)|$. $(c)$ is by Assumption 3. $(d)$ is due to $|\log(1 + x)| \leq |x|, \forall x \geq 0$. $(e)$ is due to the fact that the loss function $l$ is non-negative, i.e., $l \geq 0$. Now by applying Hoeffding's inequality (Lemma 7), we have

$$\mathbb{P}(|\mathbb{E}_{z \sim \mathsf{P}_n^o} \left[ \exp\left(l(z; \theta)\right)/\lambda \right] - \mathbb{E}_{z \sim \mathsf{P}^o} \left[ \exp\left(l(z; \theta)\right)/\lambda \right]| \geq \epsilon) \leq 2\exp\left( -\frac{2n\epsilon^2}{\exp\left(L/\underline{\lambda}\right)} \right).$$

By choosing $\epsilon = \sqrt{\frac{\exp(L/\underline{\lambda}) \log(2/\delta)}{2n}}$, we have the result. □

We prove a strong convexity result similar to Lemma 11 for KLDPO loss function.

**Lemma 14** (Strong convexity of KLDPO loss)**.** *Let $l(z; \theta)$ be the DPO loss function. The KL distributionally robust DPO loss function,*

$$\mathcal{L}^{\mathrm{KL}}(\theta; \rho) := \sup_{\mathsf{P}:\, D_{\mathrm{KL}}(\mathsf{P} \,\|\, \mathsf{P}^o) \leq \rho} \mathbb{E}_{z \sim \mathsf{P}}[l(z; \theta)],$$

*is $\gamma\lambda$-strongly convex in $\theta$ with respect to (non-weighted) 2-norm $\|\cdot\|_2$, where $\lambda$ is the regularity condition number defined in Assumption 2, and $\gamma = \frac{\beta^2 e^{4\beta B}}{(1 + e^{4\beta B})^2}$.*

*Proof.* Let $\alpha \in [0, 1]$ and $\theta, \theta' \in \Theta$. First, we denote $h(\theta; \mathsf{P}) = \mathbb{E}_{z \sim \mathsf{P}}[l(z; \theta)]$ for any $\mathsf{P}$ in the KL ball. In Lemma 10, we proved that $h$ is $\gamma$-strongly convex in $\theta$ w.r.t. norm $\|\cdot\|_{\Sigma_\mathsf{P}}$. Now observe that

$$\mathcal{L}^{\mathrm{KL}}(\alpha\theta + (1-\alpha)\theta'; \rho) = \sup_{\mathsf{P}:\, D_{\mathrm{KL}}(\mathsf{P} \,\|\, \mathsf{P}^o) \leq \rho} h(\alpha\theta + (1-\alpha)\theta'; z)$$

$$\overset{(a)}{\leq} \sup_{\mathsf{P}:\, D_{\mathrm{KL}}(\mathsf{P} \,\|\, \mathsf{P}^o) \leq \rho} \left\{ \alpha h(\theta; \mathsf{P}) + (1-\alpha)h(\theta'; \mathsf{P}) - \frac{\gamma}{2}\alpha(1-\alpha)\|\theta - \theta'\|_{\Sigma_\mathsf{P}}^2 \right\}$$

$$\overset{(b)}{\leq} \alpha\mathcal{L}^{\mathrm{KL}}(\theta; \rho) + (1-\alpha)\mathcal{L}^{\mathrm{KL}}(\theta'; \rho) + \sup_{\mathsf{P}:\, D_{\mathrm{KL}}(\mathsf{P} \,\|\, \mathsf{P}^o) \leq \rho} -\frac{\gamma}{2}\alpha(1-\alpha)\|\theta - \theta'\|_{\Sigma_\mathsf{P}}^2$$

$$= \alpha\mathcal{L}^{\mathrm{KL}}(\theta; \rho) + (1-\alpha)\mathcal{L}^{\mathrm{KL}}(\theta'; \rho) - \frac{\gamma}{2}\alpha(1-\alpha) \inf_{\mathsf{P}:\, D_{\mathrm{KL}}(\mathsf{P} \,\|\, \mathsf{P}^o) \leq \rho} \|\theta - \theta'\|_{\Sigma_\mathsf{P}}^2$$

$$\leq \alpha\mathcal{L}^{\mathrm{KL}}(\theta; \rho) + (1-\alpha)\mathcal{L}^{\mathrm{KL}}(\theta'; \rho) - \frac{\gamma}{2}\alpha(1-\alpha) \inf_{\mathsf{P}:\, D_{\mathrm{KL}}(\mathsf{P} \,\|\, \mathsf{P}^o) \leq \rho} \lambda_{\min}(\Sigma_\mathsf{P})\|\theta - \theta'\|_2^2$$

$$\overset{(c)}{\leq} \alpha\mathcal{L}^{\mathrm{KL}}(\theta; \rho) + (1-\alpha)\mathcal{L}^{\mathrm{KL}}(\theta'; \rho) - \frac{\gamma\lambda}{2}\alpha(1-\alpha)\|\theta - \theta'\|_2^2.$$

Note that the function $g(\theta) = \mathbb{E}_{z \sim \mathsf{P}}[l(z; \theta)]$ is $\gamma$-strongly convex with respect to $\|\cdot\|_{\Sigma_\mathsf{P}}$ by Lemma 10. We use this fact in $(a)$. The inequality in $(b)$ is due to $\sup_x(f(x) + g(x)) \leq \sup_x f(x) + \sup_x g(x)$. The last inequality $(c)$ is because $\lambda_{\min}(\Sigma_\mathsf{P}) \geq \lambda$, for all $\mathsf{P} \in \mathcal{P}_{\mathrm{KL}}$ by Assumption 2. This implies that $\mathcal{L}^{\mathrm{KL}}$ is a $\gamma\lambda$-strongly convex function with respect to $\|\cdot\|_2$. $\qquad \square$

### C.1 Proof of Policy Parameter Convergence of KLDPO

By Lemma 13, we have that, with probability at least $1 - \delta$,

$$\mathcal{L}^{\mathrm{KL}}(\theta_n^{\mathrm{KL}}; \rho) - \mathcal{L}^{\mathrm{KL}}(\theta^{\mathrm{KL}}; \rho)$$
$$= \mathcal{L}^{\mathrm{KL}}(\theta_n^{\mathrm{KL}}; \rho) - \mathcal{L}_n^{\mathrm{KL}}(\theta_n^{\mathrm{KL}}; \rho) + \mathcal{L}_n^{\mathrm{KL}}(\theta_n^{\mathrm{KL}}; \rho) - \mathcal{L}_n^{\mathrm{KL}}(\theta^{\mathrm{KL}}; \rho) + \mathcal{L}_n^{\mathrm{KL}}(\theta^{\mathrm{KL}}; \rho) - \mathcal{L}^{\mathrm{KL}}(\theta^{\mathrm{KL}}; \rho)$$
$$\leq |\mathcal{L}^{\mathrm{KL}}(\theta_n^{\mathrm{KL}}; \rho) - \mathcal{L}_n^{\mathrm{KL}}(\theta_n^{\mathrm{KL}}; \rho)| + |\mathcal{L}_n^{\mathrm{KL}}(\theta^{\mathrm{KL}}; \rho) - \mathcal{L}^{\mathrm{KL}}(\theta^{\mathrm{KL}}; \rho)|$$
$$\leq 2\bar{\lambda}\sqrt{\frac{\exp(L/\underline{\lambda})\log(2/\delta)}{2n}}, \quad \forall \epsilon > 0,$$

where the first inequality is because $\theta_n^{\mathrm{KL}}$ is the minimizer of $\mathcal{L}_n^{\mathrm{KL}}$. Now by the $\gamma\lambda$-strong convexity of $\mathcal{L}^{\mathrm{KL}}$ (see Lemma 14) and Lemma 5.II, we have that

$$\|\theta_n^{\mathrm{KL}} - \theta^{\mathrm{KL}}\|_2^2 \leq \sqrt{\frac{8\bar{\lambda}^2 \exp(L/\underline{\lambda})\log(2/\delta)}{\gamma^2\lambda^2 n}}, \quad \forall \epsilon > 0.$$

## D Proof of Tractable KLDPO

Next, we prove the formal version of Proposition 2.

**Theorem 3.** *Suppose we have the following distributionally robust loss that corresponds to a KL uncertainty set:*

$$\sup_{\mathsf{P}:\, D_{\mathrm{KL}}(\mathsf{P} \,\|\, \mathsf{P}_n^o) \leq \rho} \mathbb{E}_{z \sim \mathsf{P}}[l(z; \theta)].$$

*A worst distribution $\underline{\mathsf{P}} \in \mathbb{R}^n$ is related to the empirical nominal distribution $\mathsf{P}_n^o$, which is constructed using $n$ i.i.d. samples $z_1, \ldots, z_n$, through*

$$\underline{\mathsf{P}}(i) = \mathsf{P}_n^o(i) \cdot \exp\left(\frac{l(z_i; \theta) - \mu - \lambda}{\lambda}\right), \tag{23}$$

*where $\underline{\mathsf{P}}(i)$ corresponds to the worst-case mass on the $i$-th data, and further it is subject to*

$$\sum_{i=1}^n \mathsf{P}_n^o(i) \cdot \exp\left(\frac{l(z_i; \theta) - \mu - \lambda}{\lambda}\right) \cdot \left(\frac{l(z_i; \theta) - \mu - \lambda}{\lambda}\right) = \rho, \tag{24}$$

$$\sum_{i=1}^n \mathsf{P}_n^o(i) \cdot \exp\left(\frac{l(z_i; \theta) - \mu - \lambda}{\lambda}\right) = 1, \tag{25}$$

$$\lambda \geq 0. \tag{26}$$

*Proof.* We re-write the objective as a convex optimization problem

$$\underset{p \in \mathbb{R}^n}{\textbf{maximize}} \quad \langle p, \, l \rangle \tag{P1}$$

$$\textbf{subject to} \quad \sum_{i=1}^{n} p_i \log\left(\frac{p_i}{q_i}\right) \leq \rho,$$

$$\mathbf{1}^{\top} p = 1,$$

$$p_i \geq 0, \forall i.$$

First, we ignore the constraint $p_i \geq 0$ which will be automatically satisfied later. Now, the associated Lagrangian function takes the form

$$L(p, \lambda, \mu) = \langle p, \, l \rangle + \lambda(\rho - \sum_{i=1}^{n} p_i \log(p_i/q_i)) + \mu(1 - \mathbf{1}^{\top} p).$$

We can calculate the KKT conditions as follows

$$\frac{\partial L}{\partial p_i} = l_i - \lambda(\log(p_i/q_i) + 1) - \mu = 0.$$

This implies that

$$p_i = q_i \exp\left(\frac{l_i - \mu - \lambda}{\lambda}\right), \quad \forall i \in \{1, \ldots, n\}.$$

In addition, we have other KKT conditions as follows

$$\sum_{i=1}^{n} p_i \log(p_i/q_i) - \rho \leq 0,$$

$$\sum_{i=1}^{n} p_i = 1,$$

$$\lambda \geq 0,$$

$$\lambda(\sum_{i=1}^{n} p_i \log(p_i/q_i) - \rho) = 0.$$

From complimentary slackness, we have

$$\sum_{i=1}^{n} p_i \log(p_i/q_i) = \rho.$$

The unconstrained optimum would lie at a vertex far from $q$, thus the best achievable objective under the KL constraint is obtained by pushing the distribution as far as possible, thereby maximizing the utility of the KL budget. Plugging in $p_i = q_i \exp\left(\lambda^{-1}(l_i - \mu - \lambda)\right)$, we have

$$\sum_{i=1}^{n} q_i \exp\left(\frac{l_i - \mu - \lambda}{\lambda}\right) \cdot \left(\frac{l_i - \mu - \lambda}{\lambda}\right) = \rho.$$

Also, we have $\sum_{i=1}^{n} q_i \exp\left(\lambda^{-1}(l_i - \mu - \lambda)\right) = 1$. In addition, it is easy to see that the constraints $p_i \geq 0, \forall i$, are satisfied since $q_i \exp\left(\lambda^{-1}(l_i - \mu - \lambda)\right) \geq 0$. $\qquad \square$

Here, $\mu$ and $\lambda$ are implicitly defined by the constraints (Eq. (24)-Eq. (26)). Now we prove that the dual variables $-\mu - \lambda$ can be upper bounded by $-\sum_{i=1}^{n} q_i l(z_i; \theta)$.

**Proposition 4.** $-\mu - \lambda$ *satisfies* $-\mu - \lambda \leq -\sum_{i=1}^{n} \mathsf{P}_n^o(i) l(z_i; \theta)$.

*Proof.* Recall the constraint

$$\sum_{i=1}^{n} q_i \exp\left(\frac{l(z_i; \theta) - \mu - \lambda}{\lambda}\right) = 1.$$

By applying Jensen's inequality, we have

$$\exp\left(\sum_{i=1}^{n} q_i\left(\frac{l(z_i;\theta) - \mu - \lambda}{\lambda}\right)\right) \le 1.$$

Some algebra give us

$$\exp\left(\sum_{i=1}^{n} q_i\left(\frac{l(z_i;\theta)}{\lambda}\right)\right) \le \exp\left(\frac{\mu + \lambda}{\lambda}\right).$$

This implies that $-\mu - \lambda \le -\sum_{i=1}^{n} q_i l(z_i;\theta)$. $\qquad\square$

# E    Additional Experiment Results

## E.1    ArmoRM Multi-objective Alignment

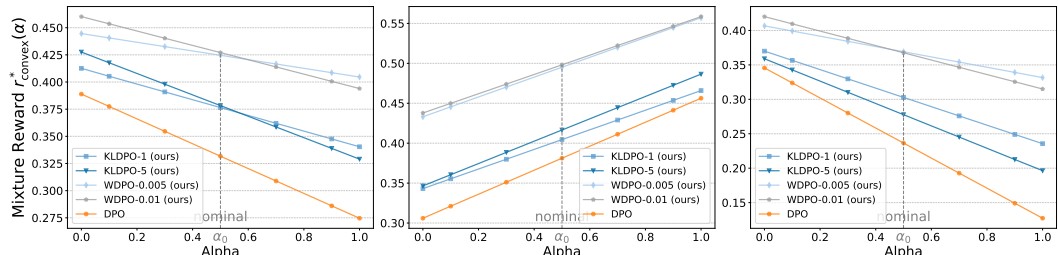

Figure 4: *Evaluation of WDPO, KLDPO and DPO on* $r^*_{\text{convex}}(\alpha)$ *in ArmoRM Multi-objective Alignment.* We evaluate WDPO with robustness parameter $\rho_o \in \{0.005, 0.01\}$ and KLDPO with robustness temperature $\tau \in \{1, 5\}$.

Similar to the Emotion Alignment experiments, we generate preference labels according to convex combinations of two reward objectives, i.e., $r^*_{\text{convex}}$ defined as *Mixture Evaluation* in previous section. Specifically, we consider **three** pairs of objectives: (1) *Ultrafeedback-Honesty* and *Helpsteer-Complexity*, (2) *Ultrafeedback-Helpfulness* and *Helpsteer-Coherence*, and (3) *Ultrafeedback-truthfulness* and *Helpsteer-Complexity*. We generate preference labels according to $\alpha^o = 0.5$ for all three cases. All models are trained for 4 epochs. Then we introduce reward shift by evaluating WDPO, KLDPO, and DPO on $r^*_{\text{convex}}(\alpha)$, where $\alpha \in \{0, 0.1, 0.3, 0.5, 0.7, 0.9, 1\}$. In the first plot of Fig. 4, the training preferences are generated accoding to reward pair (1). We observe that WDPO and KLDPO achieve superior performance compared to DPO. In particular, when reward distribution shift happens in two directions (towards standalone *Ultrafeedback-Honesty* and *Helpsteer-Complexity*), they clearly outperform DPO. In the middle plot, the training preferences are generated according to reward pair (2). We observe that WDPO with both $\rho_o = 0.005$ and $\rho_o = 0.01$ are particularly robust against reward distribution shift. Lastly, in the third plot, the training preferences are generated according reward pair (3). We observe that both WDPO and KLDPO achieve notable robustness compared to DPO.

## E.2    Leaderboard Alignment

In this section, we include alignment results evaluated on all OpenLLM Leaderboard v2 (Fourrier et al., 2024) sub-tasks. We list all sub-task names in Table 2.

**LLaMA-3.2-1B results:** In Table 3, we compare DPO, KLDPO, and WDPO trained using LLaMA-3.2-1B on **all** 39 sub-tasks of OpenLLM Leaderboard v2. We observe that our WDPO and KLDPO methods achieve superior alignment performance on the majority of subtasks. Although WDPO and KLDPO slightly underperform on few subtasks, their primary strength lies in generalization, precisely because they consistently enhance performance across a diverse range of subtasks.

**LLaMA-3.1-8B results:** In Table 4, we compare DPO and KLDPO, both trained using LLaMA-3.1-8B. Earlier, we demonstrated that WDPO, trained on LLaMA-3.2-1B, outperforms both DPO and KLDPO. However, WDPO's requirement for dual gradient computations increases its computational complexity. Due to resource constraints, we present KLDPO results for the 8B model, as it is more scalable. Following the LLaMA-3.2-1B experiments, we train KLDPO for two epochs, the point where DPO achieved optimal robustness via early stopping. Notably, KLDPO exhibits exceptional performance on math-related tasks.

Table 2: *All sub-task names in OpenLLM Leaderboard v2.*

| | | | |
|---|---|---|---|
| 1 | bbh-boolean-expressions | 21 | bbh-tracking-shuffled-objects-five-objects |
| 2 | bbh-causal-judgement | 22 | bbh-tracking-shuffled-objects-seven-objects |
| 3 | bbh-date-understanding | 23 | bbh-tracking-shuffled-objects-three-objects |
| 4 | bbh-disambiguation-qa | 24 | bbh-web-of-lies |
| 5 | bbh-formal-fallacies | 25 | gpqa-diamond |
| 6 | bbh-geometric-shapes | 26 | gpqa-extended |
| 7 | bbh-hyperbaton | 27 | gpqa-main |
| 8 | bbh-logical-deduction-five-objects | 28 | ifeval |
| 9 | bbh-logical-deduction-seven-objects | 29 | math-algebra-hard |
| 10 | bbh-logical-deduction-three-objects | 30 | math-counting-and-prob-hard |
| 11 | bbh-movie-recommendation | 31 | math-geometry-hard |
| 12 | bbh-navigate | 32 | math-intermediate-algebra-hard |
| 13 | bbh-object-counting | 33 | math-num-theory-hard |
| 14 | bbh-penguins-in-a-table | 34 | math-prealgebra-hard |
| 15 | bbh-reasoning-about-colored-objects | 35 | math-precalculus-hard |
| 16 | bbh-ruin-names | 36 | mmlu-pro |
| 17 | bbh-salient-translation-error-detection | 37 | musr-murder-mysteries |
| 18 | bbh-snarks | 38 | musr-object-placements |
| 19 | bbh-sports-understanding | 39 | musr-team-allocation |
| 20 | bbh-temporal-sequences | | |

| **LLaMA-3.2-1B** | 1 | 2 | 3 | 4 | 5 | 6 | 7 | 8 | 9 | 10 | 11 | 12 | 13 |
|---|---|---|---|---|---|---|---|---|---|---|---|---|---|
| DPO at Epoch 2 (early stopping) | 0.64 | 0.50 | 0.36 | 0.40 | 0.52 | 0.31 | 0.51 | 0.22 | 0.16 | 0.32 | 0.35 | 0.49 | 0.36 |
| DPO at Epoch 4 (goodfit) | 0.50 | 0.49 | 0.34 | 0.36 | 0.52 | 0.31 | 0.52 | 0.22 | 0.13 | 0.31 | 0.34 | 0.45 | 0.36 |
| KLDPO $\tau = 0.1$ | 0.69 | 0.52 | 0.40 | 0.43 | 0.53 | 0.34 | 0.52 | 0.18 | 0.18 | 0.33 | 0.36 | 0.48 | 0.40 |
| KLDPO $\tau = 0.05$ | 0.71 | 0.52 | 0.38 | 0.38 | 0.53 | 0.34 | 0.52 | 0.19 | 0.17 | 0.33 | 0.36 | 0.48 | 0.41 |
| WDPO $\rho_o = 0.01$ | 0.73 | 0.52 | 0.45 | 0.40 | 0.53 | 0.22 | 0.52 | 0.21 | 0.20 | 0.35 | 0.34 | 0.46 | 0.38 |
| WDPO $\rho_o = 0.005$ | 0.69 | 0.51 | 0.41 | 0.40 | 0.54 | 0.32 | 0.52 | 0.19 | 0.15 | 0.32 | 0.35 | 0.50 | 0.40 |

| **LLaMA-3.2-1B** | 14 | 15 | 16 | 17 | 18 | 19 | 20 | 21 | 22 | 23 | 24 | 25 | 26 |
|---|---|---|---|---|---|---|---|---|---|---|---|---|---|
| DPO at Epoch 2 (early stopping) | 0.25 | 0.16 | 0.14 | 0.22 | 0.53 | 0.51 | 0.23 | 0.18 | 0.17 | 0.36 | 0.49 | 0.30 | 0.28 |
| DPO at Epoch 4 (goodfit) | 0.23 | 0.15 | 0.12 | 0.22 | 0.53 | 0.49 | 0.23 | 0.20 | 0.18 | 0.35 | 0.49 | 0.30 | 0.25 |
| KLDPO $\tau = 0.1$ | 0.29 | 0.18 | 0.12 | 0.22 | 0.54 | 0.51 | 0.20 | 0.21 | 0.14 | 0.36 | 0.49 | 0.28 | 0.24 |
| KLDPO $\tau = 0.05$ | 0.29 | 0.16 | 0.12 | 0.23 | 0.54 | 0.50 | 0.20 | 0.20 | 0.14 | 0.36 | 0.49 | 0.29 | 0.25 |
| WDPO $\rho_o = 0.01$ | 0.26 | 0.18 | 0.10 | 0.30 | 0.54 | 0.50 | 0.20 | 0.18 | 0.13 | 0.39 | 0.49 | 0.24 | 0.26 |
| WDPO $\rho_o = 0.005$ | 0.26 | 0.14 | 0.12 | 0.26 | 0.54 | 0.50 | 0.16 | 0.20 | 0.16 | 0.36 | 0.49 | 0.25 | 0.26 |

| **LLaMA-3.2-1B** | 27 | 28 | 29 | 30 | 31 | 32 | 33 | 34 | 35 | 36 | 37 | 38 | 39 |
|---|---|---|---|---|---|---|---|---|---|---|---|---|---|
| DPO at Epoch 2 (early stopping) | 0.22 | 0.48 | 0.14 | 0.09 | 0.08 | 0.01 | 0.04 | 0.19 | 0.01 | 0.17 | 0.50 | 0.26 | 0.28 |
| DPO at Epoch 4 (goodfit) | 0.23 | 0.48 | 0.18 | 0.02 | 0.03 | 0.01 | 0.05 | 0.13 | 0.05 | 0.17 | 0.49 | 0.26 | 0.23 |
| KLDPO $\tau = 0.1$ | 0.24 | 0.53 | 0.20 | 0.05 | 0.08 | 0.01 | 0.08 | 0.12 | 0.02 | 0.18 | 0.52 | 0.23 | 0.24 |
| KLDPO $\tau = 0.05$ | 0.24 | 0.56 | 0.21 | 0.05 | 0.05 | 0.03 | 0.05 | 0.13 | 0.04 | 0.18 | 0.52 | 0.21 | 0.24 |
| WDPO $\rho_o = 0.01$ | 0.24 | 0.52 | 0.23 | 0.07 | 0.07 | 0.03 | 0.05 | 0.13 | 0.04 | 0.19 | 0.52 | 0.24 | 0.25 |
| WDPO $\rho_o = 0.005$ | 0.25 | 0.49 | 0.19 | 0.07 | 0.08 | 0.04 | 0.06 | 0.19 | 0.03 | 0.19 | 0.51 | 0.25 | 0.24 |

Table 3: *Evaluation of DPO, KLDPO, and WDPO on all OpenLLM Leaderboard v2 sub-tasks.*

# F   Additional Experiment Details

**Reward Model Training:** The raw Emotion dataset (Saravia et al., 2018) consists of text samples paired with multi-class labels for six different emotion classes (*joy, sadness, love, anger, fear, and surprise*). This dataset was then transformed into a multi-label dataset, referred to as the Emotion Reward Dataset. To create the multi-label dataset, the *surprise* class was excluded due to its limited representation in the original dataset. Following this, up to three random text samples from the raw dataset were concatenated, and their associated labels were merged. **This pre-processing step ensured that the reward model encountered text samples representing multiple emotions during training**.

For the reward model, GPT-2 was employed, augmented with a classification head applied to the last token. The model was trained using a sigmoid activation function and binary cross-entropy loss, adhering to the standard multilabel classification framework. Training was conducted over 8 epochs with a batch size of 64, utilizing the Adam optimizer with a learning rate of $5.0 \times 10^{-5}$ and a weight decay of 0.01. The reward model achieved a test accuracy of 84% and a test ROC-AUC score of 0.99. The emotion-specific scores predicted by this reward model were treated as the rewards for individual emotions. The ArmoRM setups did not need any reward model training.

**Supervised Fine-Tuning:** Before training the WDPO algorithm, it is essential to ensure that the model familiarize with the types of texts present in the dataset. To achieve this, we performed

| LLaMA-3.1-8B | 1 | 2 | 3 | 4 | 5 | 6 | 7 | 8 | 9 | 10 | 11 | 12 | 13 |
|---|---|---|---|---|---|---|---|---|---|---|---|---|---|
| DPO at Epoch 2 (early stopping) | 0.72 | 0.60 | 0.51 | 0.64 | 0.57 | 0.29 | 0.65 | 0.41 | 0.39 | 0.62 | 0.48 | 0.66 | 0.32 |
| DPO at Epoch 4 (goodfit) | 0.70 | 0.59 | 0.47 | 0.59 | 0.56 | 0.30 | 0.65 | 0.42 | 0.40 | 0.61 | 0.46 | 0.66 | 0.32 |
| KLDPO $\tau = 0.005$ | 0.79 | 0.58 | 0.51 | 0.61 | 0.56 | 0.33 | 0.62 | 0.35 | 0.36 | 0.63 | 0.48 | 0.66 | 0.31 |
| KLDPO $\tau = 0.01$ | 0.80 | 0.59 | 0.51 | 0.59 | 0.55 | 0.34 | 0.62 | 0.36 | 0.37 | 0.63 | 0.49 | 0.66 | 0.30 |

| LLaMA-3.1-8B | 14 | 15 | 16 | 17 | 18 | 19 | 20 | 21 | 22 | 23 | 24 | 25 | 26 |
|---|---|---|---|---|---|---|---|---|---|---|---|---|---|
| DPO at Epoch 2 (early stopping) | 0.46 | 0.66 | 0.65 | 0.51 | 0.61 | 0.68 | 0.41 | 0.21 | 0.23 | 0.34 | 0.50 | 0.30 | 0.28 |
| DPO at Epoch 4 (goodfit) | 0.47 | 0.59 | 0.66 | 0.51 | 0.61 | 0.70 | 0.40 | 0.21 | 0.20 | 0.32 | 0.50 | 0.27 | 0.31 |
| KLDPO $\tau = 0.005$ | 0.47 | 0.66 | 0.65 | 0.54 | 0.63 | 0.72 | 0.46 | 0.27 | 0.25 | 0.34 | 0.49 | 0.29 | 0.25 |
| KLDPO $\tau = 0.01$ | 0.47 | 0.65 | 0.64 | 0.53 | 0.65 | 0.74 | 0.47 | 0.28 | 0.25 | 0.33 | 0.50 | 0.32 | 0.28 |

| LLaMA-3.1-8B | 27 | 28 | 29 | 30 | 31 | 32 | 33 | 34 | 35 | 36 | 37 | 38 | 39 |
|---|---|---|---|---|---|---|---|---|---|---|---|---|---|
| DPO at Epoch 2 (early stopping) | 0.29 | 0.62 | 0.04 | 0.02 | 0.02 | 0.02 | 0.05 | 0.05 | 0.04 | 0.33 | 0.56 | 0.40 | 0.35 |
| DPO at Epoch 4 (goodfit) | 0.32 | 0.53 | 0.04 | 0.01 | 0.02 | 0.01 | 0.03 | 0.03 | 0.02 | 0.33 | 0.57 | 0.40 | 0.38 |
| KLDPO $\tau = 0.005$ | 0.33 | 0.72 | 0.42 | 0.18 | 0.10 | 0.06 | 0.25 | 0.42 | 0.10 | 0.37 | 0.54 | 0.26 | 0.24 |
| KLDPO $\tau = 0.01$ | 0.33 | 0.75 | 0.44 | 0.16 | 0.09 | 0.04 | 0.14 | 0.41 | 0.07 | 0.37 | 0.54 | 0.26 | 0.28 |

Table 4: *Evaluation of DPO and KLDPO on all OpenLLM Leaderboard v2 sub-tasks.*

supervised fine-tuning (SFT). We selected GPT-2 as the base language model and trained it to predict the next token based on the text samples in the emotion dataset. The maximum length of each text sample was capped at 68 tokens. The model was trained for 10 epochs with a batch size of 64. The training used the Adam optimizer (Kingma and Ba, 2014) with a learning rate of $5.0 \times 10^{-7}$ following 12 warmup steps. Additionally, a maximum gradient norm of 10 was applied to stabilize the training. The ArmoRM setups did not need any SFT as we used Intruct models which have already undergone multiple rounds of SFT and alignment.

**Data Generation:** (1) Emotion Alignment: A preference dataset was created, consisting of a chosen and a rejected completion for each prompt in the dataset. The first four tokens from each text in the emotion dataset were used as prompts. Using the SFT model, two completions were generated for each prompt. These completions were generated with a `top-k` value of 0, `top-p` of 1, and up to 64 new tokens. The completions were then evaluated using the reward model, and the chosen and rejected completions were determined based on a mixed metric derived from the predicted rewards. (2) ArmoRM multi-objective Alignment: Similar to the Emotion setup, we generated a preference datset by sampling two completions per prompt from the Helpsteer2 dataset. Each completion was sampled with a temperature of 0.7, `top-p` of 1 and up to 1024 new tokens. The prompts were also truncated to a maximum of 1024 tokens. We then fed these prompt-completion pairs to ArmoRM and used the scores from the first stage of the model as our multi-objective rewards. The chosen and rejected completions were determined based on a mixed metric derived from the predicted rewards. (3) Leaderboard Alignment: In this setup we sampled 10 completions per prompt in the Helpsteer2 dataset. Each completion was sampled with a temperature of 0.7, `top-p` of 1 and up to 1024 new tokens. We then fed these prompt-completion pairs to ArmoRM and used the scores from the second stage of the model as our reward, the completion with the maximum reward was our chosen completion while that with the minimum reward was our rejected completion.

**WDPO Implementation:** (1) In WDPO training, one of the main challenges is calculating the gradient penalty of the DPO loss with respect to the input. However, since the input is tokenized as integers, gradient cannot be directly calculated. To address this, gradient is calculated with respect to the output of the embedding layer, where gradients are tracked. (2) In line 4 of the tractable WDPO algorithm (Algorithm 1), we compute the gradient regularizer: $\mathcal{R}(\pi_\theta; \mathcal{D}) = \rho_o(\mathbb{E}_{z \sim \mathcal{D}} \|\nabla_z l(z; \theta)\|_2^2)^{1/2}$. A key implementation challenge arises in distributed LLM training. A naive approach computes the gradient of the pointwise DPO loss with respect to each input, averages the gradient norms over the micro-batch, and applies this as a regularizer to the batch DPO loss on each worker. However, due to the typically small micro-batch sizes in large-scale LLM training, this averaging is performed over very few samples, resulting in a highly noisy and unstable gradient penalty. To mitigate this, we exploit the inequality $\sqrt{x} \leq x$ for $x \geq 1$, allowing us to upper bound the regularizer as:

$$\mathcal{R}(\pi_\theta; \mathcal{D}) = \rho_o(\mathbb{E}_{z \sim \mathcal{D}} \|\nabla_z l(z; \theta)\|_2^2)^{1/2} \leq \rho_o(\mathbb{E}_{z \sim \mathcal{D}} \|\nabla_z l(z; \theta)\|_2^2).$$

This leads to a tractable approximation of the pointwise WDPO loss:

$$l^W(z_i, \rho_o) = l(z_i; \theta) + \rho_o \|\nabla_z l(z; \theta)\|_2^2,$$

where $l(z_i; \theta)$ denotes the standard DPO loss for sample $z_i$.

**WDPO Training:** (1) Emotion alignment: The model was trained for 40 epochs with an effective batch size of 64. We used Adam optimizer, with a learning rate of $5.0 \times 10^{-7}$ following 12 warmup steps. A maximum gradient norm of 10 was applied to ensure stable training. The DPO beta parameter was set to 0.1 for all training runs. Experiments were conducted on a single 40 GB A100 GPU, requiring gradient accumulation over two steps. (2) LLaMA experiments: The models were trained for 8 epochs with an effective batch size of 128. We used Adam optimizer with a learning rate of $5.0 \times 10^{-7}$ after 10% warmup ratio and then the learning rate was reduced using a cosine scheduler. The DPO beta parameter was set to 0.01 for all training runs. Experiments were conducted on an 8xH100 GPU setup, requiring loading the model in bfloat16 and training with DeepSpeed ZeRO-2 optimizer (Rajbhandari et al., 2020).

**KLDPO Implementation:** In line 3 of the tractable KLDPO algorithm (Algorithm 2), we compute the approximate worst-case kernel $\underline{P}(i) \propto \exp\left((1/\tau)(l(z_i;\theta) - (1/n)\sum_{i=1}^n l(z_i;\theta))\right)$. A key implementation challenge arises in distributed LLM training. A naive approach would calculate the $(1/n)\sum_{i=1}^n l(z_i;\theta)$ term by averaging $l(z_i;\theta)$ across all samples in the micro-batch of its respective worker. However, because micro-batch sizes are typically small in large-scale LLM training, this results in averaging over only a few samples, making the worst-case kernel highly noisy. To mitigate this, we introduce a synchronization step that performs an all-gather operation to collect $l(z_i;\theta)$ values from all workers. This enables averaging over the full batch across all workers, significantly reducing the noise in the worst-case kernel.

**KLDPO Training:** (1) Emotion alignment: The model was trained for 40 epochs with an effective batch size of 64. We used Adam optimizer (Kingma and Ba, 2014), with a learning rate of $5.0 \times 10^{-7}$ following 12 warmup steps. A maximum gradient norm of 10 was applied to ensure stable training. The DPO beta parameter was set to 0.1 for all training runs. Experiments were conducted on a single 40 GB A100 GPU and gradient was accumulated over two steps to keep training consistent across all algorithms. (2) LLaMA experiments: The models were trained for 8 epochs with an effective batch size of 128. We used Adam optimizer with a learning rate of $5.0 \times 10^{-7}$ after 10% warmup ratio and then the learning rate was reduced using a cosine scheduler. The DPO beta parameter was set to 0.01 for all training runs. Experiments were conducted on an 8xH100 GPU setup, requiring loading the model in bfloat16 and training with DeepSpeed ZeRO-2 optimizer (Rajbhandari et al., 2020).

# G   Limitations

**Theoretical Limitations:** Our theoretical analysis relies on Assumption 2, which ensures sufficient data coverage to guarantee strong convexity conditions. Although such data-coverage assumptions are standard within offline learning or fixed-dataset scenarios, they are moderately restrictive, as they require the training dataset to sufficiently cover the space of feature differences between the preferred and dis-preferred actions. The log-linear policy class assumption, while standard and easily extendable to neural network policies under mild additional conditions, does not constitute a significant limitation.

**Experimental Limitations:** Empirically, Wasserstein Direct Preference Optimization (WDPO) involves two separate gradient computations during training, one for calculating the gradient penalty and another for updating policy parameters via standard gradient descent. This dual-gradient requirement can increase computational complexity and training difficulty, potentially limiting practical scalability and efficiency compared to methods with a single gradient computation.

# H   Impact Statement

This paper aims to advance the field of machine learning by improving the robustness of direct preference optimization against preference model shifts. Our theoretical insights and empirical evaluations contribute to the reliability of preference-based learning methods. While our work has broad implications for AI alignment and deployment, we do not foresee any immediate societal concerns that require specific highlighting.

