# OpenReview forum: "Robust LLM Alignment via Distributionally Robust Direct Preference Optimization"
_NeurIPS.cc/2025/Conference — NeurIPS 2025 poster_

### Official Review · Reviewer_Nk1b · 2025-07-02

**Clarity:** 3
**Significance:** 3
**Originality:** 2
**Rating:** 4
**Confidence:** 4

**Summary:**

This paper proposes distributionally robust direct preference optimization approach by optimizing the distributionally robust DPO objective induced by Wasserstein distance and KL divergence, leading to W-DPO and KL-DPO. Theoretical analysis has been performed that  bounded the error of the policy parameters with high probability. The paper also details the practical implementation for W-DPO and KL-DPO and experiments on emotion alignment, multi-objective alignment, and openllm leaderboard demosntrate the efficacy of the proposed approach.

**Questions:**

Q1. In Table 1 there are two rows of DPO at Epoch 2 (early stopping) for LLaMA-3.2-1B but the number on BBH is different (0.33 vs 0.35). What leads to this difference?

Q2. Are there any in-depth analyses and ablation studies on the effect of the parameters $\tau$ and $\rho$?

Q3. This paper claims to target on distributional robustness of the algorithm. How is this property related to the benchmark settings in the experiment section? In other words, how are the metrics there connected with distributional robustness?

**Ethical Concerns:**

["NO or VERY MINOR ethics concerns only"]

**Final Justification:**

The authors are strongly encouraged to verify the numbers reported to avoid any typos and errors for the results. Other than this, the paper is in much better shape given the additional clarifications and results in the rebuttal.

**Limitations:**

yes

**Quality:**

3

**Strengths And Weaknesses:**

## Strengths

1. This paper is well-motivated by the distributionally robust optimization problem and claims to be the first to analyze for direct preference optimization.

2. The experiments covers multiple scenarios and show consistent improvement.

3. The proposed approach is clean and easy to follow.

## Weaknesses

1. Missing some in-depth analyses of the hyperparameters. See Q2 for more details.

2. The experimental evaluation protocol lacks detailed backup and justification. Please refer to Q3.

---

> ### Author Rebuttal · Authors · 2025-07-29
>
> We sincerely thank Reviewer Nk1b for the thoughtful and constructive review. We are glad to hear that you found our paper to be *“well-motivated by the distributionally robust optimization problem.”* We are also encouraged by your recognition that *“the proposed approach is clean and easy to follow”* and that *“the experiments cover multiple scenarios and show consistent improvement.”* We hope our responses and new results address your concerns and strengthen the overall contribution of the paper.
>
> **Q1**: *In Table 1 there are two rows of DPO at Epoch 2 (early stopping) for LLaMA-3.2-1B but the number on BBH is different (0.33 vs 0.35). What leads to this difference?*
>
> We thank Reviewer Nk1b for catching this typographical error. Both entries refer to the same model checkpoint, and the correct BBH score is 0.35. As verification, Table 3 in the appendix reports all 39 subtask scores on the OpenLLM Leaderboard v2, and their average confirms the 0.35 value. We will correct this in the final version.
>
> **Q2**: *Are there any in-depth analyses and ablation studies on the effect of the parameters $\rho$ and $\tau$?*
>
> We thank the reviewer for raising this important point. As discussed in **Appendix E**, we performed extensive hyperparameter analysis in both the *ArmoRM Multi-Objective Alignment* and *OpenLLM Leaderboard v2* experiments. For instance, **Figure 4** and **Table 3** (Appendix E) show that both **WDPO** and **KLDPO** exhibit relatively low sensitivity to their respective hyperparameters across tasks.
>
> To further investigate, we conducted an additional ablation using the checkpoints from **Figure 3**, where preferences were generated from  `0.5 × Ultrafeedback-Truthfulness + 0.5 × Helpsteer-Complexity`. Specifically, we evaluated:
>
> - **KLDPO** with $\tau$ = 0.5, 1, 5
> - **WDPO** with $\rho$ = 0.01, 0.005, 0.001
>
> on **500 HelpSteer2 prompts**, generating **5 completions per prompt**, and scoring all completions using **armoRM’s 19 sub-rewards**.
>
> The results confirm our earlier findings: **KLDPO** and **WDPO** demonstrate robustness across a wide range of hyperparameter values. While tuning $\tau$ and $\rho$ can further improve robustness in specific applications, both methods remain stable and effective without exhaustive tuning.
>
> | Model     | helpsteer-helpfulness | helpsteer-correctness | helpsteer-coherence | helpsteer-complexity | helpsteer-verbosity | ultrafeedback-overall_score | ultrafeedback-instruction_following |
> |:----------|:----------------------|:----------------------|:--------------------|:---------------------|:--------------------|:-----------------------------|:------------------------------------|
> | dpo       | 0.7049                | 0.7012                | 0.7645              | 0.5332               | 0.6243              | 0.6396                       | 0.7164                              |
> | kldpo-0.5 | 0.7052                | 0.7019                | 0.7653              | 0.5303               | 0.618               | 0.6402                       | 0.7162                              |
> | kldpo-1   | 0.7045                | 0.7009                | 0.7652              | 0.5316               | 0.6227              | 0.6400                       | 0.7151                              |
> | kldpo-5   | 0.7040                | 0.7003                | 0.7643              | 0.5329               | 0.6230              | 0.6391                       | 0.7144                              |
> | wdpo-d001 | 0.7088                | 0.7054                | 0.7683              | 0.5319               | 0.6269              | 0.6440                       | 0.7193                              |
> | wdpo-d005 | **0.7111**            | **0.7072**            | **0.7698**          | 0.5356               | 0.6297              | **0.6458**                   | **0.7224**                          |
> | wdpo-d01  | 0.7069                | 0.7027                | 0.7673              | **0.5368**           | **0.6298**          | 0.6417                       | 0.7183                              |
>
> | Model     | ultrafeedback-truthfulness | ultrafeedback-honesty | ultrafeedback-helpfulness | beavertails-is_safe | prometheus-score | argilla-overall_quality |
> |:----------|:---------------------------|:-----------------------|:---------------------------|:--------------------|:------------------|:-------------------------|
> | dpo       | 0.7158                     | 0.7231                 | 0.7076                     | 0.9315              | 0.7109            | 0.5397                   |
> | kldpo-0.5 | 0.7188                     | 0.7237                 | 0.7057                     | 0.9320              | 0.7086            | 0.5441                   |
> | kldpo-1   | 0.7166                     | 0.7223                 | 0.7070                     | 0.9354              | 0.7098            | 0.5400                   |
> | kldpo-5   | 0.7141                     | 0.7215                 | 0.7070                     | 0.9326              | 0.7089            | 0.5377                   |
> | wdpo-d001 | 0.7226                     | 0.7279                 | 0.7122                     | 0.9331              | 0.7185            | 0.5447                   |
> | wdpo-d005 | **0.7245**                 | **0.7303**             | **0.7171**                 | 0.9317              | **0.7225**        | **0.5490**               |
> | wdpo-d01  | 0.7170                     | 0.7250                 | 0.7139                     | **0.9365**          | 0.7206            | 0.5408                   |
>
> | Model     | argilla-judge_lm | code-complexity | code-style | code-explanation | code-instruction_following | code-readability |
> |:----------|:------------------|:----------------|:-----------|:------------------|:-----------------------------|:------------------|
> | dpo       | 0.5587            | 0.5561          | 0.5806     | 0.5538            | 0.5566                        | 0.6068            |
> | kldpo-0.5 | 0.5628            | 0.5554          | 0.5792     | 0.5526            | 0.5568                        | 0.6057            |
> | kldpo-1   | 0.5590            | 0.5554          | 0.5799     | 0.5529            | 0.5559                        | 0.6059            |
> | kldpo-5   | 0.5577            | 0.5553          | 0.5797     | 0.5534            | 0.5559                        | 0.6062            |
> | wdpo-d001 | 0.5642            | 0.5600          | 0.5834     | 0.5564            | 0.5597                        | 0.6109            |
> | wdpo-d005 | **0.5679**        | **0.5632**      | **0.5879** | **0.5607**        | **0.5632**                    | **0.6155**        |
> | wdpo-d01  | 0.5598            | 0.5619          | 0.5854     | 0.5597            | 0.5605                        | 0.6129            |
>
> **Q3**: *This paper claims to target on distributional robustness of the algorithm. How is this property related to the benchmark settings in the experiment section? In other words, how are the metrics there connected with distributional robustness?*
>
> We thank the reviewer for this important question. Our experimental design specifically targets varying degrees of preference distribution shift to evaluate distributional robustness. We summarize our approach below:
>
> 1. **Emotion Dataset (Toy Example)** – This controlled experiment introduces explicit distributional shift by varying the convex mixing coefficient between two competing objectives (e.g., “anger” vs. “fear”). It demonstrates how traditional DPO degrades under even modest shifts.
>
> 2. **ArmoRM Multi-Objective Alignment** – This more realistic setup simulates partial coverage of reward space. Models are trained on a mixture of two ArmoRM subrewards and then evaluated on other subrewards unseen during training. This mimics real-world scenarios where deployed models encounter user preferences outside their training signal.
>
> 3. **OpenLLM Leaderboard v2** – This is our most challenging and practical testbed. Models are trained using ArmoRM (ranked #32 on RewardBench, then evaluated on OpenLLM Leaderboard v2, which aggregates feedback from various human-preference and judge models. This highlights a genuine failure mode: a developer finetunes a model using ArmoRM but sees performance drop on broader evaluation. Our methods aim to mitigate this degradation.
>
> Across all settings, **distributional shift is either explicitly modeled or naturally induced**. The metric of interest in each case is the model’s reward under out-of-training-distribution preferences. Hence, **distributional robustness** is defined as the ability to maintain performance across such shifts, whether synthetic or real-world (e.g., varying mixtures of user goals, cultural expectations, or evaluation benchmarks).
>
> We believe this progression from toy to real-world tasks demonstrates the practical importance of robust preference optimization.

---

> > ### Comment · Reviewer_Nk1b · 2025-08-05
> >
> > Thank you for the detailed response and additional results. The authors are encouraged to check in detail to avoid those typos and numerical mistakes which would really affect the credibility of the paper. Furthermore, the ablations and the discussions on how the evaluation metrics are connected with distributional robustness should be clearly stated, preferably in the main text since they are critical factors to support the claims in the paper. In light of all these points, I maintain my current score of 4.

---

> > > ### Author Response · Authors · 2025-08-05
> > >
> > > We thank the reviewer for their attentive reading and for thoughtfully pointing out the one typo in the table. This level of care reflects a strong sense of responsibility, which we genuinely appreciate. We have since conducted a thorough audit of our manuscript and confirmed the consistency of all reported results across the main text and supplementary material. In particular, Table 3 in the appendix presents all 39 subtask scores from the OpenLLM Leaderboard v2 for each model. This comprehensive breakdown was included as part of our broader commitment to openness and completeness, ensuring that the evaluation results are not only accurate but also reported with a high degree of granularity.
> > >
> > > Regarding the request for clearer discussion of ablations and their connection to distributional robustness, we appreciate this important suggestion. While our original submission included ablation studies in the appendix, we have extended them further during the rebuttal. Due to space limitations, we were unable to incorporate all such results in the main text, but we agree that highlighting how evaluation metrics reflect distributional robustness would strengthen the narrative. We will revise the final version accordingly.
> > >
> > > We sincerely thank the reviewer again for their thoughtful and constructive feedback.

---

### Official Review · Reviewer_oGJc · 2025-07-02

**Clarity:** 3
**Significance:** 3
**Originality:** 3
**Rating:** 4
**Confidence:** 4

**Summary:**

This paper suggests novel preference alignment algorithms, WDPO and KLDPO, which provides preference distribution robust learning against distribution shifts that occur between training and testing phases.

**Questions:**

1. I would like to know how much the performance differs on common benchmarks such as AlpacaEval 2.0 or Arena Hard when using widely adopted preference alignment datasets like UltraFeedback and comparing against algorithms such as DPO or Dr. DPO

**Ethical Concerns:**

["NO or VERY MINOR ethics concerns only"]

**Final Justification:**

My core concern was the absence of a baseline and the lower performance of the trained model on the Leaderboard compared to SFT. However, the baseline has been added, and the lower performance was due to different reward criteria, so my concerns have been resolved.

**Limitations:**

Yes

**Quality:**

2

**Strengths And Weaknesses:**

Strengths
1. Preference distribution shift is an important problem and is particularly critical because in real-world scenarios, the preference distributions between training and testing phases rarely align perfectly. This methodology effectively addresses this issue.
2. These algorithms are general-purpose methods that can be directly applied without requiring specific information about the preference distributions during training and testing.
3. The experiments were conducted across various distribution shift setups and evaluated on multiple models.

Weaknesses
1. This research is closely related to robust preference learning study (such as IPO and Dr. DPO). However, this study lacks baseline experiments and only compares against DPO.
2. The experimental performance on Leaderboard Alignment is questionable. For LLaMA-3.2-1B, DPO performance mostly decreases compared to SFT, and while both KLDPO and WDPO show smaller performance drops (almost similar) than SFT, this does not constitute meaningful performance gains. Wouldn't this indicate a flaw in experimental setup?
3. While preference distribution shift is indeed a problem, it is not as extreme as the current experimental setup suggests. Switching objectives between anger and fear for emotional alignment represents an overly dramatic scenario. Figure 2 shows that in convex mixing, DPO outperforms even at alpha 0.3~0.4. Although DPO's performance drops significantly at higher alpha values, I question whether such setups represent common situations. Alpha 0-0.4 range better represents typical distribution shift scenarios, and DPO performs better in this range. These experimental results suggest that KLDPO and WDPO actually decrease performance compared to DPO when distribution shifts are minimal, which undermines the utility of these methods.

---

> ### Author Rebuttal · Authors · 2025-07-29
>
> We thank Reviewer oGJc for taking the time to carefully review our work and for providing thoughtful and detailed feedback. We are glad to know that the reviewer found that *"preference distribution shift is an important problem"*, *"[our] methodology effectively addresses this issue"*, and *"experiments were conducted across various distribution shift setups and evaluated on multiple models"*. Below, we address your specific concerns.
>
> **Q1**: *Comparison between SFT and DPO.*
>
> We thank Reviewer oGJc for raising a critical point regarding the observation that DPO underperforms the SFT model (LLaMA-3.2-1B-Instruct) on the OpenLLM Leaderboard v2.
>
> **To clarify upfront**: when evaluated on the *same* reward model used to generate the preference dataset, armoRM, **DPO consistently outperforms the SFT baseline**, as expected. We verified this through a controlled evaluation (see supplementary tables below), using the exact model checkpoints from Table 1, and scoring model outputs across all 19 subrewards with armoRM. This confirms the experimental pipeline is sound and not flawed. As expected from classical ML, longer training improves in-distribution performance on the training reward.
>
> | Model                      | helpsteer-helpfulness | helpsteer-correctness | helpsteer-coherence | helpsteer-complexity | helpsteer-verbosity | ultrafeedback-overall_score | ultrafeedback-instruction_following |
> |----------------------------|------------------------|------------------------|----------------------|----------------------|----------------------|------------------------------|--------------------------------------|
> | DPO at Epoch 2             | 0.7256                 | 0.7210                 | **0.7848**           | 0.5245               | 0.5967               | 0.6637                       | 0.7449                               |
> | DPO at Epoch 4             | **0.7286**             | **0.7238**             | **0.7848**           | **0.5233**           | **0.6000**           | **0.6682**                   | **0.7529**                           |
> | LLaMA-3.2-1B-Instruct (SFT)| 0.6588                 | 0.6603                 | 0.7415               | 0.4904               | 0.5598               | 0.6031                       | 0.6521                               |
>
>
> | Model                      | ultrafeedback-truthfulness | ultrafeedback-honesty | ultrafeedback-helpfulness | beavertails-is_safe | prometheus-score | argilla-overall_quality |
> |----------------------------|-----------------------------|------------------------|----------------------------|---------------------|------------------|--------------------------|
> | DPO at Epoch 2             | 0.7505                      | 0.7550                 | 0.7269                     | 0.9281              | 0.7637           | 0.5798                   |
> | DPO at Epoch 4             | **0.7571**                  | **0.7661**             | **0.7362**                 | **0.9345**          | **0.7707**       | **0.5840**               |
> | LLaMA-3.2-1B-Instruct (SFT)| 0.6823                      | 0.6749                 | 0.6342                     | 0.9309              | 0.6342           | 0.5156                   |
>
>
> | Model                      | argilla-judge_lm | code-complexity | code-style | code-explanation | code-instruction_following | code-readability |
> |----------------------------|------------------|------------------|-------------|-------------------|----------------------------|-------------------|
> | DPO at Epoch 2             | 0.5980           | 0.5764           | 0.5954      | 0.5715            | 0.5827                     | 0.6294            |
> | DPO at Epoch 4             | **0.6028**       | **0.5792**       | **0.5971**  | **0.5746**        | **0.5851**                 | **0.6319**        |
> | LLaMA-3.2-1B-Instruct (SFT)| 0.5211           | 0.5157           | 0.5386      | 0.5077            | 0.5258                     | 0.5618            |
>
> We sampled 500 test prompts from the HelpSteer 2 dataset. For each model, we generated five completions per prompt. These were then scored using armoRM across 19 subrewards, aggregated across 2500 completions.
>
>
> Now back to Table 1 in our paper, the underperformance of DPO stems from a well-known issue in RLHF: **reward hacking and distributional overfitting**. As studied in Azar et al. (2024) and other works on reward hacking (Amodei et al., 2016; Skalse et al., 2022; Eisenstein et al., 2023), methods like DPO (and RLHF PPO) can overfit to the training-time reward model, here, armoRM, when the test-time reward or preference distribution shifts, e.g., to OpenLLM Leaderboard v2.
>
> We hope this resolves the concern and reinforces the importance of developing preference optimization methods that are robust to reward and preference distribution shifts.
>
>
> **Q2**: *...this study lacks baseline experiments and only compares against DPO...*
>
>
> We thank Reviewer oGJc for highlighting the importance of comparing against other robust preference learning methods such as IPO and DRDPO. In response, we trained IPO and DRDPO using preferences generated according to 0.5 × Ultrafeedback-Truthfulness + 0.5 × Helpsteer-Complexity. Following our evaluation protocol, we sampled 500 prompts from the HelpSteer 2 test set and generated five completions per model per prompt. Our results show that both KLDPO and WDPO outperform DPO and IPO on several key sub-rewards under the armoRM evaluation. While DRDPO performs competitively on some tasks, KLDPO and WDPO consistently achieve stronger overall scores, confirming the benefit of our proposed distributionally robust methods.
> | Model     | HelpSteer-Coherence | HelpSteer-Correctness | HelpSteer-Helpfulness | UltraFeedback-Honesty | UltraFeedback-Instruction_Following | UltraFeedback-Truthfulness | UltraFeedback-Overall Score |
> |-----------|---------------------|------------------------|------------------------|------------------------|--------------------------------------|-----------------------------|------------------------------|
> | DPO       | 0.7645              | 0.7012                 | 0.7049                 | 0.7231                 | 0.7164                               | 0.7158                      | 0.6396                       |
> | IPO       | 0.7615              | 0.6967                 | 0.6995                 | 0.7165                 | 0.7088                               | 0.7121                      | 0.6356                       |
> | DRDPO     | 0.7626              | 0.6981                 | 0.7029                 | 0.7230                 | 0.7137                               | 0.7122                      | 0.6393                       |
> | KLDPO     | 0.7653              | 0.7019                 | 0.7052                 | 0.7237                 | 0.7162                               | 0.7188                      | 0.6402                       |
> | WDPO      | **0.7698**          | **0.7072**             | **0.7111**             | **0.7303**             | **0.7224**                           | **0.7245**                  | **0.6458**                   |
>
>
> **Q3**: *While preference distribution shift is indeed a problem, it is not as extreme as the current experimental setup suggests. Switching objectives between anger and fear for emotional alignment represents an overly dramatic scenario.*
>
> We appreciate Reviewer oGJc’s perspective on the severity of preference distribution shifts in our experimental setup. While we agree that the emotion alignment experiment is a simplified toy example designed to isolate and illustrate core failure modes, our remaining experiments address far more realistic and impactful scenarios.
>
> In particular, our second experiment uses **armoRM**, a strong, publicly available reward model that currently ranks **#32** on RewardBench, outperforming many proprietary models. For comparison:
>
> - **google/gemini-2.5-pro** scores **#30**
> - **openai/gpt-4.1-mini-2025-04-14** scores **#33**
>
> Notably, **Arena-Hard 2**, a widely used benchmark for LLM evaluation, includes models such as **Gemini-2.5**, **Gemini-2.5-Flash**, and **OpenChat 4**, which are in a similar performance tier. armoRM also surpasses GPT-4 in LLM-as-a-Judge setups, making it a viable and meaningful alternative for annotation and benchmarking tasks (e.g., MT-Bench, AlpacaEval-2.0, Arena-Hard).
>
> In our third experiment, we evaluate models on the **OpenLLM Leaderboard v2**, which aggregates scores across multiple axes and reflects the state-of-the-art in community-driven LLM evaluation. These setups represent genuine real-world scenarios where preference shift arises, such as switching reward models between training, tuning, and deployment phases.
>
> Thus, our proposed methods address not only extreme preference shifts but also moderate and subtle ones that are common yet hard to detect in practice. Their value lies in mitigating performance degradation across this full spectrum of shift severity, which is central to ensuring robust alignment in deployed systems.
>
> **References**
>
> Mohammad Gheshlaghi Azar, Mark Rowland, Bilal Piot, Daniel Guo, Daniele Calandriello, Michal Valko, and Rémi Munos. A general theoretical paradigm to understand learning from human preferences. ArXiv, abs/2310.12036, 2023
>
> Amodei, D., Olah, C., Steinhardt, J., Christiano, P., Schulman, J., and Mané, D. (2016). Concrete problems in ai safety. arXiv preprint arXiv:1606.06565.
>
> Skalse, J., Howe, N., Krasheninnikov, D., and Krueger, D. (2022). Defining and characterizing reward gaming. Advances in Neural Information Processing Systems, 35:9460–9471.

---

> > ### Author Response · Authors · 2025-08-05
> >
> > We sincerely thank Reviewer oGJc for the time and care taken in providing a thoughtful and detailed review of our work. Your questions and comments, particularly regarding baseline comparisons, experimental setups, and evaluation on common benchmarks, have been highly valuable in refining and clarifying our contributions. In our rebuttal, we have provided detailed responses along with additional experiments to address each of your concerns. We would be grateful to know whether these clarifications and results satisfactorily resolve the issues you raised.

---

> > > ### Comment · Reviewer_oGJc · 2025-08-06
> > >
> > > Thank you for your detailed response. Most of my concerns have been addressed, and I'll raise score to 4

---

### Official Review · Reviewer_4S36 · 2025-07-03

**Clarity:** 3
**Significance:** 3
**Originality:** 3
**Rating:** 5
**Confidence:** 2

**Summary:**

In this paper, the authors introduce a new family of DPO-style losses call Robust-DPO that (loosely speaking) optimize for the robustness of models across data distributions. The particular problem they motivate is well illustrated in Figure 1: vanilla DPO often catastropically fails when exposed to new distributions of data, which are commonly encountered given the non-homogenous nature of preference data. Their approach is based on modeling directly (what they call) an "uncertainty set" of data distributions around a nominal distribution (e.g., the training distribution), which they use to  perform minimax-style optimization (intuitively: the goal is to minimize the expected loss for each instance wrt the distribution in the uncertainty set that maximizes the loss of that instance, as given in Eq. 6). While I am not familiar with minimax optimization, I found their description very easy to follow, as well as their technical description of why such optimization is difficult.

Operationally, they define two variants of Robust-DPO, Wasserstein DPO (WDPO, Eq. 7) and Kullback-Leibler DPO (KLDPO, Eq. 8) that use Wassertein and KL metrics, respectively, to parameterize these uncertainty sets. Rather impressively, they prove finite-sample error bounds for both (Theorem 1 and Theorem 2), yet due to the difficult of the mini-max optimization, devise tractable variants of both (Section 6) that facilitate SGD-style optimization (i.e., the SGD algorithms they define in Algorithms 1-2).

For experiments, they perform (somewhat contrived)  on-policy style offline learning using a variety of medium size llama models (1B->8B) using the HelpSteer2 dataset and perform evaluation on the more diverse OpenLLM leaderboard (from what I understand, the OpenLLM represents a more distributionally diverse test set, though these decisions need to be better explained and motivated in the text). Their main results are in Figure 2-3 and Table 1, where their distributionally robust variants achieve significant improvements over the vanilla DPO.

**Questions:**

-- Did you train on datasets besides HelpSteer? If so, why not?

-- It seems like your approach still assumes that data is of high quality. Do you think that your approach could be extended for modeling data with noise? A discussion of this in the paper would be helpful.

**Ethical Concerns:**

["NO or VERY MINOR ethics concerns only"]

**Final Justification:**

I found that the author(s) addressed all my concerns. Providing a Pytorch sketch of the implementation also helped tremendously to clarity certain things. I therefore have decided to keep my already high score high.

**Limitations:**

yes

**Quality:**

3

**Strengths And Weaknesses:**

# strengths

-- A very novel distributionally robust variant of DPO and objective that aims to improve DPO's robustness to distribution shifts (an important challenge that is well motivated in the paper) based on minimax-style optimization. I can imagine such work influential otehr work in this area.

-- Deep theoretical results about error bounds of two specific losses they derive (WDPO and KLDPO), coupled with tractable variants of each loss and novel SGD-style algorithms that effectively solve the corresponding minimax optimization problems.

-- Empirical evidence of the effectiveness and utility of their approach over vanilla DPO.

# weaknesses

-- As noted above, the experiments, while sufficient for showing the utility of their approach, are fairly contrived. Given that most standard preference datasets, such as ultrafeedback, are actually aggregations of different datasets (and hence, different distributions of data), I would suspect that their approach would be quite useful in the standard setting too. No such results are reported (see questions below).

-- (minor) When technically very impressive and comprehensive, it is unclear how easily their variants of SGD (Algorithms 1-2) would fit into existing DPO training libraries, which could limit the impact of these new approaches. I would like to see the authors address this. A code example would be very useful to have in the appendix.

---

> ### Author Rebuttal · Authors · 2025-07-31
>
> We sincerely thank Reviewer 4S36 for their thoughtful and encouraging review. We are especially grateful that you found our formulation of distributionally robust preference optimization to be *“very novel”* and *“well motivated,”* and that you appreciated the theoretical depth of our work, including the error bounds and SGD-style algorithms. We are also pleased that you found our empirical evidence convincing and the paper clearly written, despite the technical complexity. Below, we address your insightful questions and suggestions, particularly regarding dataset choices, robustness to noisy data, and integration into existing DPO training frameworks.
>
> **Q1**: *When technically very impressive and comprehensive, it is unclear how easily their variants of SGD (Algorithms 1-2) would fit into existing DPO training libraries, which could limit the impact of these new approaches. I would like to see the authors address this. A code example would be very useful to have in the appendix.*
>
> We thank the reviewer for highlighting the importance of integration with existing DPO pipelines. Our implementations of **WDPO** and **KLDPO** are designed for minimal disruption: we modify only the loss function, keeping the rest of the DPO pipeline (data, sampling, training loop, etc.) unchanged.
>
> To illustrate this, we provide the following pseudocode sketch built directly upon the `DPOLoss` structure in `openrlhf`. The core DPO logic remains intact and is reused in both `KLDPOLoss` and `WDPOLoss`. Only the lines marked between `===` blocks are additions specific to each variant.
>
> ```python
> class (KL/W)DPOLoss(nn.Module):
>     def __init__(self, ...) -> None:
>         super().__init__()
>         # DPO parameters
>         ...
>
>         # KLDPO and WDPO hyperparameters
>         self.kldpo_tau = ...
>         self.wdpo_rho = ...
>
>     def forward(
>         self,
>         policy_chosen_logps, policy_rejected_logps,
>         reference_chosen_logps, reference_rejected_logps,
>         ...
>     ) -> Tuple[torch.Tensor, torch.Tensor, torch.Tensor]:
>
>         # Calculating vanilla DPO loss (common to DPO, KLDPOLoss, and WDPOLoss)
>         pi_logratios = policy_chosen_logps - policy_rejected_logps
>         ref_logratios = reference_chosen_logps - reference_rejected_logps
>         logits = pi_logratios - ref_logratios
>
>         losses = (
>             -F.logsigmoid(self.beta * logits) * (1 - self.label_smoothing)
>             - F.logsigmoid(-self.beta * logits) * self.label_smoothing
>         )
>
>         # ======== KLDPO Extension (added in KLDPOLoss) ========
>         all_losses = [torch.zeros_like(losses) for _ in range(dist.get_world_size())]
>         dist.all_gather(all_losses, losses.detach())
>         all_losses = torch.cat(all_losses)  # Gather losses across all distributed ranks
>
>         kl_reweights = torch.exp((losses - all_losses.mean()) / self.kldpo_tau).clamp(max=10).detach()
>         # KL reweighting approximates the adversarial distribution in KL-DRO
>
>         loss = (kl_reweights * losses).mean()  # Final KLDPO objective
>         # =======================================================
>
>         # ======== WDPO Extension (added in WDPOLoss) ========
>         grads = torch.autograd.grad(
>             outputs=losses,
>             inputs=model_embeddings,
>             grad_outputs=torch.ones_like(losses),
>             create_graph=(train_eval == 'train')
>         )[0]
>         # Calculate WDPO gradient penalty
>
>         grads = grads.pow(2).sum(dim=(-1, -2))
>         grads = grads[: len(grads) // 2] + grads[len(grads) // 2 :]
>         losses = losses + self.wdpo_rho * grads
>         loss = losses.mean()  # Final WDPO objective
>         # ===================================================
>
>         chosen_rewards = self.beta * (policy_chosen_logps - reference_chosen_logps).detach()
>         rejected_rewards = self.beta * (policy_rejected_logps - reference_rejected_logps).detach()
>
>         return loss, chosen_rewards, rejected_rewards
> ```
> **Q2**: *Did you train on datasets besides HelpSteer? If so, why not?*
>
>
> Yes, we use two distinct prompt sources in our experiments. The distinction we study is not just prompt diversity, but **preference distribution shift**, so we fix the prompt pool (e.g., HelpSteer2) and vary how reward signals are assigned.
>
> | Experiment                          | Prompt Source       | Preference / Reward Source                     | Why this Setup?                                                                 |
> |-------------------------------------|----------------------|------------------------------------------------|----------------------------------------------------------------------------------|
> | **Toy (Fig 2)**                     | *Emotion Dataset*    | Synthetic pairwise labels (e.g., “anger” vs. “fear”) | Controlled shift between two competing objectives to test DPO degradation.      |
> | **ArmoRM Multi-Objective (Fig 3)**  | **HelpSteer2** (~20k prompts) | *ArmoRM* sub-objectives (Llama3-8B model)       | Allows structured composition of sub-rewards for training, then evaluation on sub-objectives **unseen during training**. |
> | **OpenLLM Leaderboard v2 (Table 1)**| **HelpSteer2**       | Evaluation via OpenLLM leaderboard (GPT-4, Claude, etc.) | Trained on ArmoRM preferences, evaluated using out-of-distribution judge models. |
>
> ---
>
> **Why not UltraFeedback prompts?**
>
> While *UltraFeedback* has 60k prompts, its pairwise preference labels are fixed and cannot be flexibly recomposed. In contrast, *HelpSteer2* prompts combined with **ArmoRM** enable structured and interpretable simulations of preference shift.
>
> ---
>
> **What is ArmoRM and why use it?**
>
> ArmoRM is **not a black-box chatbot judge** like GPT-4 or Claude. It is a **multi-objective reward model** trained with *absolute ratings across human-aligned dimensions* (e.g., honesty, helpfulness, verbosity). This design allows us to explicitly control which sub-objectives are used for training vs. testing, precisely modeling the kind of distribution shift our methods are built to handle.
>
> **Q3**: *It seems like your approach still assumes that data is of high quality. Do you think that your approach could be extended for modeling data with noise? A discussion of this in the paper would be helpful.*
>
>
> We thank the reviewer for raising this important point. While our methods were not originally designed to explicitly model noisy preference labels, we explored this direction in a follow-up experiment and observed promising robustness trends.
>
> Specifically, we trained DPO, KLDPO, and WDPO on preferences generated by a synthetic reward:
> `0.5 × HelpSteer-Complexity + 0.5 × UltraFeedback-Truthfulness`,
> and then introduced **10\%** and **20\% label noise** by randomly flipping a subset of the pairwise preferences.
>
> All models were trained on these corrupted datasets and evaluated on 500 prompts from the HelpSteer2 test set, each with 5 completions. These completions were scored using the same two ArmoRM subrewards: `HelpSteer-Complexity` and `UltraFeedback-Truthfulness`, to isolate the effect of label noise as the only varying factor.
>
>
> The results below show that WDPO consistently maintains performance under increasing noise, especially compared to DPO and KLDPO:
>
>
> | Model         | \| 0% Noise \|                  |                            | \| 10% Noise \|                         |                            | \| 20% Noise \|                         |                            |
> |---------------|----------------------------------|----------------------------|------------------------------------------|----------------------------|------------------------------------------|----------------------------|
> |               | Helpsteer-Complexity             | Ultrafeedback-Truthfulness | Helpsteer-Complexity                     | Ultrafeedback-Truthfulness | Helpsteer-Complexity                     | Ultrafeedback-Truthfulness |
> | dpo           | 0.5332                           | 0.7158                     | 0.5293                                   | 0.7140                     | 0.5245                                   | 0.7135                     |
> | kldpo-5       | 0.5329                           | 0.7141                     | 0.5287                                   | 0.7154                     | 0.5238                                   | 0.7124                     |
> | wdpo-d001     | 0.5319                           | 0.7226                 | **0.5293**                               | **0.7197**                 | **0.5250**                               | **0.7172**                 |
>
> This suggests that our distributionally robust formulation, though originally motivated by test-time preference shifts, may also confer some resilience against noisy training data.

---

> > ### Comment · Reviewer_4S36 · 2025-08-06
> >
> > Thank you for all of these clarifications and for the detailed rebuttal.
> >
> > I find the code in particular to be very helpful, please consider putting this into the camera ready, it really helps to clarify key details.
> >
> > I will continue to keep my high score and note that the authors addressed all of my concerns is a satisfactory manner.

---

### Official Review · Reviewer_S16R · 2025-07-03

**Clarity:** 3
**Significance:** 3
**Originality:** 3
**Rating:** 5
**Confidence:** 3

**Summary:**

This paper introduces two distributionally robust extensions of Direct Preference Optimization (DPO), named **Wasserstein DPO (WDPO)** and **Kullback-Leibler DPO (KLDPO)**, to address alignment failures of large language models under preference distribution shifts. The authors develop tractable gradient-based algorithms and provide theoretical guarantees. Empirical results show that WDPO and KLDPO consistently outperform standard DPO in scenarios with shifting or diverse user preferences.

**Questions:**

1. Can you show the effectiveness of your methods on larger models and larger datasets?
2. Can your distributional robustness framework be extended to non-DPO alignment methods such as PPO-based RLHF or Imitation Learning?
3. Have you considered evaluating your methods under real-world preference shift scenarios, such as regional, demographic, or language-based distributions, rather than synthetic datasets designed for simplified testing?
4. Can you discuss more about the computational overhead of your methods?

**Ethical Concerns:**

["NO or VERY MINOR ethics concerns only"]

**Final Justification:**

The paper is technically solid and insightful. The authors have also sufficiently addressed my concerns. Therefore I recommend acceptance.

**Limitations:**

Yes.

**Paper Formatting Concerns:**

No.

**Quality:**

3

**Strengths And Weaknesses:**

**Strengths**:

- Proposes practical solutions (WDPO and KLDPO) for addressing preference distribution shift, a critical issue in real-world LLM deployment.

- Combines solid theoretical guarantees with scalable, implementable algorithms.

- Empirical results consistently demonstrate robustness and improvement over standard DPO.

**Weaknesses**:

- Experimental validation is limited in scale, with relatively small models and datasets.

- Benchmark setups are relatively simple and may not fully reflect real-world diversity.

---

> ### Author Rebuttal · Authors · 2025-07-31
>
> We thank Reviewer S16R for the thoughtful and detailed review. We are encouraged to hear that you found our work to provide *“practical solutions (WDPO and KLDPO) for addressing preference distribution shift,”* which you identify as *“a critical issue in real-world LLM deployment.”* We also appreciate your positive remarks on the combination of *“solid theoretical guarantees with scalable, implementable algorithms,”* and your recognition that our *“empirical results consistently demonstrate robustness and improvement over standard DPO.”* We hope our responses further clarify the contributions and scope of our work.
>
> **Q1**: *Can you show the effectiveness of your methods on larger models and larger datasets?*
>
> We appreciate the reviewer’s interest in scaling our methods. Due to resource constraints common in academic settings, we were limited to training models up to the size of LLaMA 8B. While this may not match the scale of industry models, we believe it still provides valuable insights into scalability. Notably:
>
> - Our algorithms were applied across a range of model sizes: LLaMA 1B (used in the ArmoRM multi-objective alignment task), 3B, and 8B.
> - ArmoRM itself is a powerful LLaMA-8B reward model, offering a strong and challenging signal for preference optimization.
> - Our successful extension of KLDPO to LLaMA 8B demonstrates practical scalability within feasible academic compute budgets.
>
> We agree that further evaluation at larger scales is an important future direction, and we hope to pursue this as additional compute resources become available.
>
> **Q2**: *Can your distributional robustness framework be extended to non-DPO alignment methods such as PPO-based RLHF or Imitation Learning?*
>
> Thank you for this insightful question. Extending distributionally robust alignment beyond DPO is a compelling future direction.
>
> One promising avenue is to robustify the reward model itself, learning a **distributionally robust reward function** that generalizes well under test-time distribution shifts. For instance, in preference modeling, we can consider a reward function of the form $r_\theta(s, a) = \langle \theta,\; \phi(s, a) \rangle$, where $\phi$ is a feature map, and estimate it via maximum likelihood using a preference dataset $\mathcal{D} = \{(s\_i, a^1\_i, a^2\_i, y\_i)\}\_{i=1}^N \sim P^o$. The negative log-likelihood loss of $\theta$ is
> $$
> \mathcal{L}(\theta \mid \mathcal{D}) = - \sum\_{i=1}^N \log[\mathbf{1} (y\_i=1) \sigma( \langle {\theta,\;\phi(s_i,a_i^1)-\phi(s_i,a_i^2)}\rangle) + \mathbf{1} (y\_i=0) \sigma( \langle {\theta,\;\phi(s_i,a_i^2)-\phi(s_i,a_i^1)}\rangle)].
> $$
>
> Robustifying this objective with respect to distributional shifts in $P^o$, which is used to sample $\mathcal{D}$, could yield a reward model that is resilient to preference drift.
>
> This robust reward can in turn be used for PPO-style RLHF, as part of a broader robust alignment pipeline.
>
> We also note that recent works have proposed distributionally robust behavior cloning and imitation learning algorithms (Bashiri et al.; Panaganti et al.), which could complement our framework in future explorations.
>
> References:
>
> M. A. Bashiri, B. Ziebart, and X. Zhang, “Distributionally robust imitation learning,” Advances in neural information processing systems.
>
> Panaganti, K., Xu, Z., Kalathil, D., and Ghavamzadeh, M. (2023b). Distributionally robust behavioral cloning for robust imitation learning. In 2023 62nd IEEE Conference on Decision and Control (CDC), pages 1342-1347.vol. 34, pp. 24404–24417, 2021.
>
> **Q3**: *Have you considered evaluating your methods under real-world preference shift scenarios, such as regional, demographic, or language-based distributions, rather than synthetic datasets designed for simplified testing?*
>
> We thank the reviewer for this valuable suggestion. While we agree that regional, demographic, and language-based preference shifts represent important real-world challenges, public datasets with explicit annotations for such axes of variation remain limited or proprietary. As a result, conducting controlled and reproducible experiments along these lines is currently nontrivial.
>
> That said, we would like to emphasize that our experimental setup already captures meaningful and realistic distribution shifts:
>
> - In our **ArmoRM multi-objective alignment** experiment, we simulate preference shifts by training on a mixture of sub-objectives (e.g., helpfulness + correctness) and evaluating on sub-objectives held out during training (e.g., coherence or verbosity). This mimics real-world scenarios where a deployed model may face user goals not seen in training.
>
> - In our **OpenLLM Leaderboard evaluation**, we train using ArmoRM, which represents a single (though strong) judge model, and test on a diverse aggregation of preferences across multiple human and automated evaluation sources. This shift from training to evaluation naturally induces real-world preference mismatch, even if not explicitly labeled by region or demographic.
>
> We agree that future work could explore explicit real-world axes such as culture, age, or language, especially as more datasets with these attributes become publicly available. We are excited by this direction and view our current work as a foundation for robust preference optimization under broader forms of distributional variability.
>
> **Q4**: *Can you discuss more about the computational overhead of your methods?*
>
> We thank the reviewer for raising this important point. We provide a detailed discussion on computational overhead and implementation optimizations in **Appendix F** of our paper.
>
> Briefly:
>
> - **WDPO** introduces a gradient penalty term that requires differentiating the DPO loss w.r.t. model input. Since tokenized inputs are discrete, we track gradients at the embedding layer. Moreover, to reduce variance from small micro-batch sizes in distributed LLM training, we upper-bound the square-root penalty with its squared counterpart, leading to a tractable per-example regularizer.
>
> - **KLDPO** introduces a reweighting scheme across the full batch. To stabilize this under distributed training, we insert an `all_gather` operation to synchronize losses across workers, mitigating the high variance from local micro-batch estimates.
>
> Despite these added steps, the rest of the DPO pipeline (optimizer, learning rate schedule, model parallelism setup) remains **unchanged**. We found the overall overhead to be modest and manageable across both small-scale (single A100) and large-scale (8xH100 with DeepSpeed ZeRO-2) experiments.
>
> We hope this reassures the reviewer that our methods are practical and scalable with careful engineering, and we will make these implementation strategies explicit in our updated appendix.

---

> > ### Comment · Reviewer_S16R · 2025-08-06
> > **Thanks for the rebuttal**
> >
> > I would like to thank the authors for the rebuttal. My concerns have been adequately addressed.

---

> > > ### Author Response · Authors · 2025-08-06
> > >
> > > We sincerely thank Reviewer S16R for the thoughtful, constructive, and encouraging review. We greatly appreciate your recognition of WDPO and KLDPO as practical solutions to the critical challenge of preference distribution shift, and of our combination of solid theoretical guarantees with scalable, implementable algorithms. Your questions on scaling to larger models and datasets, extending robustness to other alignment paradigms, exploring real-world shift scenarios, and quantifying computational overhead prompted valuable clarifications and helped us better frame our contributions and future directions. We are glad to hear that our rebuttal has fully addressed your concerns, and we deeply appreciate your engagement and support throughout the review process.

---

### Note · Authors · 2025-08-11

We thank all reviewers and the AC for their time, thoughtful feedback, and constructive engagement, which have helped us improve the clarity, scope, and presentation of our work.

**Area Chair**: We sincerely appreciate your time and guidance in facilitating the review process. During rebuttal, we added baseline comparisons (IPO, DRDPO), ran controlled armoRM evaluations confirming pipeline soundness, and expanded discussion on applicability to larger models, diverse datasets, real-world shift scenarios, and computational trade-offs. Across reviews, the novelty, theoretical guarantees, and empirical robustness of our methods were recognized. WDPO and KLDPO tackle a critical, underexplored challenge in LLM alignment: maintaining performance under preference distribution shift. Our formulation unifies minimax-style robustness with scalable SGD algorithms, supported by finite-sample guarantees and practical engineering optimizations, reinforcing the validity and significance of our contributions.

**Reviewer S16R**: We are grateful for your recognition of WDPO and KLDPO as practical, theoretically grounded solutions, and for your probing questions on scaling, extending to other alignment paradigms, real-world shift scenarios, and computational overhead. Your points helped us clarify the broader applicability and engineering practicality of our methods.

**Reviewer 4S36**: Thank you for your encouraging review and clear summary. Your questions on evaluation beyond HelpSteer, inclusion of a code example, and handling noisy data led us to expand our discussion on applicability to broader datasets and future directions in noise-robust alignment.

**Reviewer oGJc**: Thank you for highlighting the importance of preference distribution shift and for raising detailed questions on baselines, experimental setups, and benchmark selection. We addressed these with new IPO and DRDPO comparisons, clarified the role of extreme vs. moderate shift scenarios, and provided controlled evaluations confirming the soundness of our pipeline.

**Reviewer Nk1b**: We appreciate your careful reading, catching the typo, and suggesting an ablation study. We extended our hyperparameter analysis with additional ablations, and clarified that our experiments, from controlled toy tasks to realistic benchmarks, capture both extreme and subtle preference shifts.

We again thank the reviewers and AC for their engagement throughout the process.

---

### Decision · Program_Chairs · 2025-09-17

**Decision:**

Accept (poster)

**Comment:**

The current paper introduces two distributionally robust versions of Direct Preference Optimization (DPO), motivated by the need to address alignment failures of large language models (LLMs) under preference distribution shift.
The reviewing panel generally appreciated the relevance of the problem addressed, which is of practical importance given the widespread deployment of LLMs in real-world applications where preference distributions may vary significantly from those seen during training.
Reviewers were also in agreement in praising the method as theoretically grounded and implemented in a way that is practically scalable.
The empirical evaluations demonstrate robust improvement over DPO across various benchmarks, and, while a raised weakness of the submission was the limited scale of the experimental validation, the rebuttals helped improving the experimental validation of the method by providing additional experiments comparing the proposed method against other robust preference learning methods.
Overall, the paper presents a well-motivated and technically sound approach to enhancing the robustness of LLM alignment, that reviewers unanimously recommended for acceptance.